# Ethnobotany around the Virovitica Area in NW Slavonia (Continental Croatia)—Record of Rare Edible Use of Fungus *Sarccoscypha coccinea*

**DOI:** 10.3390/plants13152153

**Published:** 2024-08-03

**Authors:** Ivana Vitasović-Kosić, Dominik Berec, Łukasz Łuczaj, Riccardo Motti, Josip Juračak

**Affiliations:** 1University of Zagreb Faculty of Agriculture, Svetošimunska cesta 25, 10000 Zagreb, Croatia; dberec@student.agr.hr (D.B.); jjuracak@agr.hr (J.J.); 2Institute of Biology, University of Rzeszów, ul. Zelwerowicza 8B, 35-601 Rzeszów, Poland; lukasz.luczaj@interia.pl; 3Department of Agricultural Sciences, University of Naples Federico II, Via Universita 100, 80055 Portici, Italy; motti@unina.it

**Keywords:** ethnobotanical knowledge, edible wild mushrooms, wild edible plants, medicinal plants, traditional uses of plants

## Abstract

Slavonia is the most developed agricultural region in Croatia. With rich and fertile soils that have enabled the cultivation of a wide variety of fruits, vegetables, and cereals, Slavonia has always met the food needs of its population. Today, the biocultural diversity of local varieties and semi-natural vegetation has irretrievably disappeared. Our aim was to document the remaining local knowledge of plant use in this area through in-depth semi-structured interviews, which were conducted in 2022–2023. All possible aspects of the use of plants and fungi were recorded as food, animal feed, medicine, construction, jewelry, rituals and ceremonies, dyes, etc. The names and uses of local plant varieties were also recorded. The results show 1702 entries—a total of 296 plant taxa from 76 families and 28 fungi from 16 families. The most frequently named plants were: *Urtica dioica*, *Robinia pseudoacacia*, *Rosa canina*, and *Sambucus nigra*. The plants with the greatest variety of uses were *Morus alba*, *Rosmarinus officinalis*, *Triticum aestivum*, and *Zea mays*. Interesting uses were identified. The leaves of the ornamental plant *Hosta sieboldiana* are still used today as food for wrapping meat with rice, the aquatic plant *Trapa natans* is eaten like chestnuts, and *Pteridium aquilinum* was once consumed as a vegetable. In addition, *Ambrosia artemisiifolia* and *Sambucus ebulus* were given to horses to prevent and avoid blood poisoning. Some forest species had a special significance and were revered or favored. The most frequently mentioned edible fungi were *Boletus* sp., *Cantharellus cibarius*, and *Lactarius piperatus*. *Auricularia auricula-judae* is the only species stated to have been used exclusively as a raw snack. Evidence of edible use of *Sarccoscypha coccinea*, which was reported as traditionally consumed in the past, was of particular interest. Despite the modernization and agricultural nature of the region, many interesting uses of plants and fungi were identified. Further efforts should be directed towards documenting this knowledge to facilitate its dissemination in the communities that possess it, or at least to preserve it for future generations.

## 1. Introduction

Due to its ecological and socio-economic conditions, Slavonia is the most developed agricultural area in Croatia. It has always met the food needs of its population, and its rich and fertile soils have enabled the cultivation of a wide variety of fruits, vegetables, and cereals. For this reason, the use of wild plants in the diet of the people of Slavonia is generally not greatly pronounced, as is common for peasants from Slavic countries, who used to resort to several of the most common wild greens, ignoring other species [1]. In contrast to Slavonia, the region of Dalmatia in Croatia is ethnobotanically very interesting, with the combination of Slavic and Mediterranean influence positively affecting the number of wild plants used in everyday life [2].

In the last century, the records of traditional foraging practices are derived from travelogues, historical ethnographic papers, and accounts of famines, which have impacted nearly every country on Earth [3,4,5,6]. In most European countries, wild plant use intensified in times of famine and war [3]. When the population of Slavonia underwent mass starvation, especially during Ottoman rule (16th and 17th centuries), chestnuts (*Castanea sativa* Mill.) were added to wheat flour to bake bread. Remarkably, there has been no major famine in Slavonia for about 175 years, since the mid-19th century, not even during the First and Second World Wars. The widespread use of acorns (*Quercus* sp.) as a raw material for bread making was also not recorded [7]. However, the feeding of acorns to animals was widespread and is used today in organic pig farming [8].

Due to the rich and fertile soils of the Pannonian Plain, the area of Slavonia was colonized several times in the 19th century by settlers from other parts of the former Austro-Hungarian monarchy [9]. After the First World War, during the Kingdom of Yugoslavia, and then after the Second World War, at the beginning of the former socialist Yugoslavia, the population was resettled to Slavonia from the poor regions of the country, such as Herzegovina, the Dalmatian hinterland, Bosnia and Montenegro. As a result, Slavonia saw the assimilation and mixing of Slavic and Germanic peoples and, to a lesser extent, the Hungarians who lived along the natural border of the Drava River.

The different peoples brought with them knowledge of plant cultivation and various crafts. They had little need to use wild plants as food. In the 19th and first half of the 20th century, forestry and hunting of wild animals were important economic activities in addition to agriculture [10]. All this is reflected in the traditional continental cuisine of Slavonia, which is based on pork meat and river fish, a variety of cereals, paprika, suet, browned flour, pork fat, roasts, cultivated fruits (quince, currant, gooseberry, plum, apricot, etc.) and naturalized fruits planted by the order of Empress Maria Theresa (e.g., white and black mulberry, *Morus alba* and *M. nigra*) [11,12].

In 2008 Ivan Šugar [13] published a systematic collection of Croatian plant names “Hrvatski biljni imenoslov”, which was created on the basis of the author’s many years of fieldwork on the collection of vernacular names for plants throughout Croatia. Between 1962 and 1986, the Yugoslav Army conducted a project in which Josip Bakić from the Institute for Naval Medicine of the Yugoslav Navy in Split documented traditional knowledge of wild foods and analyzed their chemical composition. He trained army personnel, developed survival tactics, organized survival expeditions, and shared the results of his studies and experiments with the general public [14,15,16,17,18,19]. Thorough systematic ethnobotanical research in Croatia started over a decade ago, on the Adriatic coast, with two authors of this paper, in the regions of Dalmatia [2,11], Primorje, Istria [20,21,22], then Inland Dalmatia [23] and Lika [24]. Continental Croatia has only been explored recently, and only partially so, e.g., around Varaždin [25,26,27]. No ethnobotanical research has been conducted in Northwest Slavonia. The closest region to have been the subject of research was Northeast Pannonia [28], where medicinal plant use was recorded.

Presently, traditional knowledge is being forgotten due to changes in lifestyle, migration, depopulation, and the transition to a “modern way of life”, without coexistence with nature. Łuczaj [29] emphasizes that biocultural biodiversity is irretrievably disappearing, and hence the main goal and absolute priority of ethnobiological research is to document disappearing traditional knowledge. Given this context, the aim of the present paper is to document and highlight the persistence of local ethnobotanical knowledge of people living in NW Slavonia, a continental part of Croatia.

## 2. Results and Discussion

In the survey conducted between July 2022 and September 2023, 1702 data records on plants known to the respondents were collected. A total of 296 plant taxa from 76 families and 28 fungi from 16 families were recorded (Table 1). Of the total of 296 plant taxa, 36 were only named by one informant.

In the interviews conducted, the average number of plant taxa mentioned per respondent was 44.6, ranging from 16 to 98 (SD = ±18.7). Fungi were mentioned in 30 surveys, with an average of 4.3 fungal species per respondent. The largest number of plant taxa belongs to the families Rosaceae (64), Vitaceae (27), and Asteraceae (20). Certain families are characterized by a high number of mentions of a single plant taxon, such as the families Urticaceae (*Urtica dioica*, 34 times) and Viburnaceae (*Sambucus nigra*, 28 times).

Among 296 plant taxa, the 5 mentioned most frequently are *Urtica dioica* (34), *Sambucus nigra* (28), *Rosa canina* (26), *Robinia pseudoacacia* (26), and *Taraxacum* spp. (25). A total of 12 out of 296 plant taxa have a frequency of more than 20, and 43 of them are mentioned more than 10 times. *Vitis vinifera*, *Malus domestica*, and *Pyrus communis* are the plant species with the most recorded local varieties (25, 19, and 10, respectively) (Table 1).

While the number of plant varieties mentioned is larger in reality, they are not listed in the table because their cultivation and use are widespread today, and no specific application was recorded. Moreover, according to the interviewees, the great genetic diversity of the varieties has been lost with the advent of seed companies.

The social, economic, and cultural importance of a plant for an area can be judged by the type and frequency of its use. Plants in the study area are used for food, beverages, cosmetics, alcoholic beverages, animal feed, tools and equipment, building and construction, ceremonial purposes, medicine, and other unspecified uses. The most common use category is food (846 entries), followed by medicine (217 entries), other unspecified uses (214 entries), and beverages (188 entries). The least frequently mentioned uses are ceremonial (48 entries), building and construction (35), and cosmetics (21). The greatest variety of plants is found in the use category food (210 plant taxa), followed by other uses (102 plant taxa) and medicine (95 plant taxa). For the purpose of data presentation, we narrowed the use categories from 10 to 6 in Figure 1.

*Urtica dioica*, *Robinia pseudoacacia*, and *Sambucus nigra* are the three taxa with the highest values in the usage reports (UR). Their use in different use categories was reported 50 times or more in the surveys (Figure 2). The taxa with the highest UR values are mainly used for food, beverages, medicinal, and other purposes (Table 2). Among them, *Taraxacum* spp. has the highest uniformity of use as measured by the fidelity level (*FL*) (*FL* = 0.96) for the food use category, followed by *Rosa canina* (*FL* = 0.79). The taxa *Tilia* spp. (*FL* = 0.86) and *Sambucus nigra* (*FL* = 0.84) have the highest degree of fidelity for the non-alcoholic beverage usage category.

For the principal component analysis (PCA), six variables or usage categories were formed, so that some usage types were combined into one:Food and non-alcoholic beverages are combined into “Food or drink”;Medical use and cosmetics are combined into “Medicine or cosmetics”;Uses for construction, making of tools and utensils have been combined into “Building, tools or utensils”;Ceremonial use and other unspecified uses are summarized under “Other uses”.

The PCA calculations are based on the absolute frequencies of plant uses in the various use categories. A PCA biplot shows the dispersion of the plants in relation to the first two components (Figure 3). The percentage of variance attributable to the first component is 60.487%, and the percentage of variance attributable to the second component is 12.438%.

The variable “Food or drink” has the highest positive loading of the first component. In the second component, the use categories “Building, tools or utensils” and “Other uses” have the highest positive loading, while the use category “Alcoholic beverage” has the highest negative loading. According to the PCA scores obtained, in the area of negative values of component 1, we find plants used for construction, manufacture of tools, and for other purposes not mentioned (Figure 3). In the second quadrant, we find plants used for the making of food or beverages, while in the third quadrant, we find edible plants and those used for the production of alcoholic beverages.

The number of interviews in which the use of a particular plant is mentioned is highest for the most frequently listed plant taxa: *Urtica dioica* (34) and *Sambucus nigra* (28), which means that all interviewees who mentioned these plants also mentioned at least one category of use for them. Relative frequency of citation (*RFC*) for these taxa is 0.900 and 0.775, respectively. The taxa *Robinia pseudoacacia* (*RFC* = 0.725) and *Rosa canina* (0.725) also have high *RFCs*. The plants with the greatest diversity of uses are *Morus alba*, *Rosmarinus officinalis*, *Triticum aestivum*, and *Zea mays*. They all belong to seven out of the ten use categories.

The importance of the taxa for the study area was assessed using the cultural importance (*CI*) and relative importance (*RI*) indices. *Urtica dioica* (*CI =* 1.625), *Robinia pseudoacacia* (*CI =* 1.275) and *Sambucus nigra* (*CI =* 1.25) have the highest cultural value for the community. Two of the plants with the highest *CI* values are also among the three plants with the highest values of *RI*. Five plants with the highest *RI* are *Urtica dioica* (*RI* = 0.929), *Sambucus nigra*, (*RI* = 0.859), *Morus alba* (*RI* = 0.847), *Robinia pseudoacacia* (*RI* = 0.831), and *Tilia* sp. (*RI* = 0.734).

The most frequently used part of the plant is the fruit, which is mentioned 153 times. This is followed by leaves, with 83 mentions; above-ground parts, with 48; and seeds, with 45 mentions. Fruits are usually eaten, while the flower and leaf parts are used for drinks, medicinal purposes, and food. The chord diagram in Figure 4 presents the 15 plants with the highest *UR* according to their most frequently utilized parts.

### 2.1. Cultivated Plants

Since Northwest Slavonia is an agricultural region, the cultivation of plants has been an integral part of the duties and traditions of the local population. In the past, households were usually self-sufficient, producing their own food in vegetable gardens, vineyards, and orchards. It was common for housewives to spend their time in the vegetable garden as a secondary task to running the household. The food produced in the gardens and fields formed the basis for feeding people and domestic animals, as was also the case in the Central Lika region [24]. Surplus food was prepared and preserved for the winter using the following methods: drying, pickling, canning, storage in pits (mostly potatoes, using a pit called “trap” in Croatian), or in attics. Fruit species are mentioned particularly frequently, with *Malus domestica*, *Vitis vinifera*, *Prunus domestica*, and *Pyrus communis* as the basis of fruit consumption and the production of strong alcoholic beverages (brandy and liqueur). The fruits of these species were regularly dried for the winter and processed into jam. The pear variety ‘Tikvica’ (*Pyrus communis* ‘Tikvica’) was most commonly dried. In Central Lika, the variety ‘Jesenka’ was used for this purpose [24].

The population of the studied area fed on cultivated fruit species from orchards and vineyards, and much less on wild fruit species. The apple tree was an indispensable fruit species, and its use was widespread, whether fresh, for pies, dried on a string or next to the oven, for making vinegar, compote, tea, brandy, juice, or baked as a dessert. We recorded a total of 19 varieties of apples.

The grapevine was an indispensable part of fruit growing. Due to their natural resistance to many diseases, the most grown grape varieties were ‘Tudum’ (*Vitis* x *labruscania*) and ‘Izabela’ (*Vitis vinifera* ‘Isabella’). People favored these varieties because they produced the best yields at a time when only blue vitriol (CuSO_4_) and Bordeaux mixture were used to protect the plants. Although the commercial production of ‘Tudum’ wine is forbidden today, the local population prefers must and wine from this variety. In the Samobor area, the local population also produced wine from the Tudum variety [26]. In the Virovitica area and the Žumberak–Somoborsko gorje [26], the varieties ‘Isabella’, ‘Frankovka’, ‘Chardonnay’, ‘Kraljevina’, ‘Muškat’, ‘Pinot’, ‘Rheinriesling’, ‘Tudum’ and ‘Welschriesling’ are cultivated, from which it can be concluded that the range of grape varieties in these areas was very similar.

The quince (*Cydonia oblonga*) was particularly prized in winter when few fresh fruits were available. It was used to make compote, jam, and quince cake: quince cheese, the local “kitnikez” (originally “Quittenkäse” in German). Quince was regularly placed on the bedroom cupboard as a room fragrance.

The walnut (*Juglans regia*) was regularly planted to celebrate the birth of a child, but also to provide shade in the gardens. The green fruits are used to make a strong alcoholic drink “orahovac”, which is also mentioned in the Lika region [24] and Žumberak–Samoborsko gorje [26]. Its ground kernels are a common ingredient in various walnut cakes (“orahnjača”). The walnut leaf was used against pests (insects) or for smoking as a tobacco substitute.

Of the other fruit varieties recorded, brandy is made from figs (*Ficus carica*) or apricots (*Prunus armeniaca*). Mulberries (*Morus* sp.) are used for “dudovača” brandy, similar to the custom in the Mediterranean part of Croatia [24].

The local name and variety of apricot “kajsija”, whose kernels taste like almonds (*Prunus amygdaloides* Schltr.), are also consumed. The apricot is gradually disappearing from the area of NW Slavonia due to late frosts.

In the 19th century, *Morus alba* and *M. nigra* were planted on a large scale in the territory of the Austro-Hungarian monarchy by order of the Austro-Hungarian Empress Maria Theresa in order to produce fodder for mulberry silk (L.), which was processed in two silk factories in the Virovitica region [12].

Laurel (*Laurus nobilis*) was an indispensable spice when pickling cabbage or preparing bean stew, and this custom was also widespread in other parts of Croatia [25].

Numerous ornamental flowers were an essential part of the well-tended garden. In addition to beautifying the garden, the flowers often had a useful value: chrysanthemums (*Chrysanthemum* × *morifolium* (Ramat.) Hemsl.) were brought to the cemetery especially for All Saints’ Day, lilies (*Lilium candidum*) were carried to the church for the feast of St. Anthony (as in the town of Varaždin [25]), and were used to produce an ointment that promoted the healing of burns. *Forsythia* (*Forsythia* sp.) and narcissus (*Narcissus pseudonarcissus* and *N. poeticus*) were also prepared and carried to church on Palm Sunday (the Sunday before Easter).

Pelargonium (*Pelargonium zonale*) was a popular flower used by girls to decorate their hair (Virovitica Municipal Museum), which is also documented for the Žumberak–Samoborsko gorje Nature Park area [26].

The leaves of the ornamental plant *Hosta sieboldiana* are still used today as food to wrap meat with rice (in a dish called “sarma”). Dogan et al. [46] recorded a few dozen species used for this purpose in SE Europe and SW Asia, but this species was not mentioned in their review and is an interesting local culinary tradition, maybe a recent innovation. *Hosta* spp. are however used in Japanese cooking [47].

*Cannabis sativa* and *Linum usitatissimum* were used for fiber production, as in the Varaždin region [25], Lika [24], and the Žumberak–Samoborsko gorje Nature Park area [26].

*Sempervivum tectorum* was planted in front of houses as protection from evil forces, or on roofs to protect houses from thunder. This belief is also present in neighboring Serbia [32], coastal parts of Croatia (ŁŁ, IVK personal observations, unpublished observations), and Poland [31].

### 2.2. Cultivated Vegetable and Cereals Taxa

Agriculture was practiced very intensively in the researched area. One of the reasons for this was the breeding of numerous domestic animals (cattle and horses), whose feed requirements had to be met. The main crops grown in the fields were: *Hordeum vulgare*, *Avena sativa*, *Linum usitatissimum*, *Triticum aestivum*, *Secale cereale*, and *Zea mays*.

The plants grown in vegetable gardens are still important today, and more recently the cultivation of *Ipomea batatas* was recorded. The interviewees state that, in the past, much more *Vicia faba* and *Pastinaca sativa*, different varieties of *Phaseolus vulgaris*, and *Pisum sativum* were grown, then “thrown away” and forgotten with the advent of modern varieties. To save space in the field, *Zea mays*, *Phaseolus vulgaris*, and *Cucurbita pepo* were often sown together, similar to the Varaždin region [25]. *Valerianella locusta* was also often sown between *Solanum tuberosum*.

### 2.3. Animal Feed and Medicine

Animals were called “treasures”, and horses were loved and appreciated because they were a means of transport (carriage) and the most important “mechanization” for work in the fields. They were also a status symbol. Sometimes they were even treated as “family members”. The “Lipicans” [48] were the most famous. Even today, there is a state stud farm for Lipicans in the town of Lipik.

The horses’ feed consisted of *Zea mays* and *Triticum vulgare* seeds, to which *Urtica dioica* was added as a “healthy plant” and a source of protein. *U. dioica* was also fed to turkeys, chickens, etc.

In addition, common ragweed (*Ambrosia artemisiifolia*) and danewort (*Sambucus ebulus*) were used to prevent disease; they were given to horses to prevent and avoid blood poisoning, called “ukrviti se”. This particular use is not known in the neighboring countries, but for comparison, in Western Herzegovina combinations of *Sambucus ebulus* juice, whole egg, oil, and soot were used against mastitis, and oak bark was used to release blood above the udder [49]. It is interesting that *A. artemisifolia*, an invasive plant from North America, became a traditional medicine for horses. Although occasionally recorded as a traditional medicinal plant in North America, its use in Europe is surprising [50]. Respondents claimed that the invasive *A. artemisiifolia* appeared during the Second World War with humanitarian aid packages from North America.

### 2.4. Wild Plants

In total, 108 taxa of useful wild plants were recorded in the study area. One possible reason for a relatively good knowledge of wild herbs is the organized buying that took place in the past in buying stations, where people brought collected herbs that were then resold to pharmaceutical factories. In this way, the local population was able to further increase their household incomes. Table 1 shows which herbs were sold (for sale).

Picking fruits or above-ground parts of shoots of perennial plants in moderate quantities is rarely problematic, except in the case of rare and endangered species whose harvest could endanger many edible and medicinal plant species [51]. In this category, the water nut (*Trapa natans*) is a nearly threatened (NT) species of Croatian flora, according to IUCN. It grows by water and is traditionally used as food, similar to chestnuts [37]. Its large seeds are edible, have a sweet taste, and are rich in starch. They are eaten boiled, fried, or ground into flour.

Many wild-growing plants are of medicinal use and cultivated in gardens. Benedictine monks, who raised the level of health culture in Croatia, grew medicinal plants in their monastic gardens, and practiced medicine, are believed to have introduced this custom [36,52].

Plants are usually used in the form of infusions or decoctions. Alternatively, they are soaked in brandy. The leaves or grated roots are placed on wounds to drain pus and facilitate the healing process. The best indicator of trust in the healing power of plants is the large quantities of fresh and dried plants available every day at open-air markets in Croatia [36].

An interesting newly reported use is the utilization of young shoots of *Phragmites australis* for snacking, flour (boiled root), building material in traditional houses, and as shelter from the sun. A similar use as a sweetener (sugar substitute) has been documented for young shoots of *Arundo donax*, used in the Izola Region (Slovenia) [33]. It was most likely introduced during sugar shortages caused by World Wars I and II or by the poorest of the poor at the time. *Pteridium aquilinum* was used as a vegetable in the past. The shoots and underground rhizomes of the species are widely used as food in Asia [53,54], but the food use of the shoots has hardly been recorded in Europe. Only in the Basque region of Spain, young fronds were occasionally chewed raw as a snack [55], or cooked in Istria (Croatia) [22]. Occasionally also bracken rhozmes were used as famine food, e.g., in Belarus [56] and France [3]. 

At a time when coffee was not on the market or was very expensive, the local population prepared a substitute for coffee, so-called “white coffee”, from the root of wild-growing chicory (*Cichorium intybus*) [22].

Some forest species had a special significance, i.e., they were revered and favored. These included many oak species (*Quercus* spp.) and the wild cherry (*Prunus avium*), whose wood was particularly valued for its hardness and color. The trunks (“majpan”) of tall hornbeam (*Carpinus betulus*), beech (*Fagus sylvatica*), poplar (*Populus* sp.), or willow (*Salix* sp.) were felled on May 1st, placed onsite when the construction of a house began, or decorated with ribbons and placed in the central squares. This custom probably spread from Central European countries [57,58,59].

On the eves and mornings of holidays, magical procedures were performed to protect family members and animals from diseases and troubles throughout the year. Plants play a significant role in these rituals because they transfer their vital and protective power to people and animals. An illustrative example is a custom of washing one’s face with water in which spring flowers (*Viola* sp., *Bellis* sp., *Taraxacum* spp., etc.), were soaked on the morning of Palm Sunday. Plants brought to church on the same day also symbolize health. *Cornus mas*, a plant of health par excellence, is the most important among them; its curative power is evident from the folk saying “healthy as a cornelian cherry” [36].

Other plants brought into the church, such as *Rosmarinus officinalis*, *Laurus nobilis*, *Taxus baccata*, *Corylus avellana*, and *Salix* sp., were also regarded as guardians of health. Their power was enhanced by the blessing of the church. In the Northern Croatian coastal region, olive and palm branches are brought to the church to be blessed [30].

The custom of using olive branches for Palm Sunday has recently (around 1980) spread from coastal parts of Croatia [30], and the blessed branches are kept until the next Palm Sunday to protect the house from bad weather. In some regions, these plants were burned as incense.

In another area of Slavonia, around Slavonski Brod, *Cornus mas*, *Salix caprea*, *Dipsacus sylvestris* Mill., *Filipendula hexapetala* Gilib., and *Epilobium parviflorum* L. are tied into a bouquet on Palm Sunday [43]. *Cornus mas* stands for general health, while the other plants are used for specific diseases. For example, the inflorescences of *Salix caprea* (“maca”) are eaten for chest pain, *Dipsacus sylvestris* is used for scabies and mange, *Epilobium parviflorum* is burned as incense for skin rashes, and *Filipendula hexapetala* heals those afflicted by evil powers [43]. The bouquets in this area are also brought for consecration on other feast days, e.g., on the feasts of St. John, St. Peter, and St. Paul. They are also kept for apotropaic purposes and to protect health.

### 2.5. Fungi

As many as 28 taxa of fungi were recorded, 27 of which are used exclusively as food. Eight were mentioned only once. Most fungi are used following heat treatment (boiling, frying). They are more rarely consumed raw. Sometimes they are preserved by drying or pickling. *Boletus* sp. (19 mentions), *Cantharellus cibarius* (16 mentions), and *Lactarius piperatus* (12 mentions) were the most commonly used (Table 1).

In comparison, *Agaricus campestris*, *Boletus edulis*, *Cantharellus cibarius*, and *Macrolepiota procera* are the most common in Northwestern Continental Croatia. In the Varaždin [25] and Central Lika regions [24], only the first three were mentioned. Only the use of *Boletus edulis* and *Cantharellus cibarius* was recorded in the Žumberak–Samoborsko gorje area [26]. Similar species to those recorded in our study area are also used in the Dalmatian Zagora [60] and other regions of Europe [57,58].

*Auricularia auricula-judae* is the only species stated to have only been used as a raw snack. The same name and use were recently recorded from Serbia [45], though there it was reported to have been used in salads or cooked in soups, and considered to improve blood vessel function. No other food use of the species has been reported as traditional in Europe.

*Fomes fomentarius* is a fungus used in beekeeping, where it is lit to smoke the bees. Its use as kindling has also been mentioned; it is particularly important because it keeps the embers burning for a long time. Interestingly, it is also used to make an infusion (hot drink).

In contrast to wild plants, which were traditionally not collected on a large scale because there was no need for them (no famine), wild mushrooms had always been collected and eaten in Slavonia on a large scale. The local population knew them and ate them without fear.

Fungi were mostly picked by men, who knew the forest better. Only known fungi were collected to avoid possible poisoning. The porcini mushroom (*Boletus* sp.) was most prized, and prepared as a sauce, breaded, or used as a spice in stews. The surplus was cut into thin slices and dried in the sun to preserve it throughout the year. Such a custom was also widespread in the Žumberak–Samoborsko gorje National Park [26] and in Varaždin [25]. For many, the sale of dried porcini mushrooms was an additional source of income.

Some fungi, such as the field fungi (*Agaricus campestris*), the honey fungi (*Armillariella* spp.), and the shield mushroom (*Entholoma clypeatum*) grew in orchards, which made them a special and easily accessible delicacy. They were prepared in goulashes.

Puffballs (Lycoperdaceae) were eaten as a raw snack or fried. They are also used in the neighboring Serbia [45].

The shaggy ink cap (*Coprinus comatus*) is not consumed with alcohol. It was suspected to contain the compound coprine, which can react with ethanol and have a toxic effect on the body. Therefore, alcohol should be avoided before, during, and after coprine consumption [61]. This view was widespread among the local mushroom pickers but finally, no coprine was found in this species [49]. In Serbia, *C. comatus* is used cooked or fried, dried, and used as chips. Medicinally, it is used for lowering blood sugar levels [45].

Evidence of the edible use of *Sarcoscypha coccinea* is of particular interest. It has hardly been reported as traditionally eaten in the past. Only in Serbia was it reported as part of a poached dessert [45]. However, the taxon has recently become popular in Europe, e.g., in the UK [62] and Poland (Łuczaj, personal observations, unpublished data) due to its interesting color and easy identification of the genus.

## 3. Materials and Methods

### 3.1. Description of the Area Studied

The Croatian region of Slavonia is not administratively defined, but in a broader sense, it includes the administrative areas of four counties (Osjek-Baranja, Požega-Slavonija, Brodsko-Posavska, and Vukovarsko-Srijemska), the largest part of Virovitica-Podravska County, and smaller parts of Bjelovar-Bilogorska and Sisak-Moslavina Counties. Thanks to its soil resources and suitable climatic conditions, Slavonia is the most important agricultural area in the Republic of Croatia, especially for agricultural production, and is often referred to as the “granary of Croatia”. The five counties that are wholly or largely part of Slavonia (out of a total of twenty-one counties in Croatia) make up 22% of the country’s total area, while their share of the total area of arable land is 58% [63]. The average agricultural area in Croatia is 7.04 ha per farm, and 14.49 ha per farm in the Slavonian counties [63].

The climatic characteristics of this area can be described as a fresh continental climate. The average annual temperature in the city is 10 °C. Air temperatures rise throughout the year and reach their maximum in July and August. The climate in this area is characterized by the fact that there are no dry periods in the year and precipitation is distributed throughout the year. The total annual rainfall is 808 mm and there are two rainfall peaks per year, the first in June and the second in November. The lowest rainfall occurs in late summer, early fall, and winter. The area is a typical lowland region and lies around 80 to 130 m above sea level [64,65].

The geological conditions and relief favor the socio-economic upgrading of the area and are not a limiting factor for development. The settlements are located in the flat part of the town and on the gentle slopes of Bilogora, which ensures uninterrupted economic utilization and infrastructure development. The Slavonian economy has always been strong. At the beginning of the 20th century, the oak forests in the entire forest area began to be exploited through investments of French capital [10]. In the Virovitica area, the timber company Tvin d.o.o. was founded in 1913 and still produces furniture today. In 1976, Viro tvornica šećera d.d., a factory for the production and processing of sugar based in Virovitica, was founded. In addition, there used to be two silk factories for the production of mulberry silk [66].

### 3.2. Data Collection

The data were collected through interviews during extensive field visits to a Slavonian municipality—the town of Virovitica and surrounding settlements (Figure 5).

The interviews were conducted in the period from July 2022 to September 2023 using the semi-structured method of in-depth interviewing and the technique of free listing. Interviewees were selected from the local population using local connections and the snowball method or based on recommendations from key interviewees. Whenever possible, the interviews were conducted outdoors so that the interviewees could not only name the plant but also see and recognize it. In addition to the fname, interviewees were asked about the plant’s uses, the parts used, and the methods used to prepare or process the plant. Thirty-five people were interviewed, with an average age of 78.37 years, ranging from 48 to 96 years. The interviewees included 22 (63%) women and 13 (37%) men. The principles of the American Anthropological Association Code of Ethics [67] and the International Society of Ethnobiology Code of Ethics [68] were followed in conducting the interviews.

For each plant, interviewees could name one or more of the following modalities or categories for different uses: food, beverage (non-alcoholic), cosmetics, alcoholic beverage, animal feed, tool or utensil, building or construction, ceremonial use, medicine, and other unspecified uses. The use reports were then used to assess their importance to the inhabitants of the study area.

Standard floras for this area of Europe were used to identify and authenticate the plants, e.g., Nikolić’s guide to the identification of the flora of Croatia [69], Pignatti’s Flora of Italy [70], and the Flora Croatica Database [71]. The plant names are compared with WFO online [72]. The voucher specimens were collected, herbarized and stored in the herbarium of the Faculty of Agriculture of the University of Zagreb, ZAGR (http://herbarium.agr.hr/ (accessed 22 October 2023)). The fungi were housed at the Faculty of Agriculture of the University of Zagreb in Zagreb, and the names of the fungi follow the Index Fungorum [73].

### 3.3. Data Analysis

The following variables from the final dataset were used to analyze the collected data: interview ID, scientific plant name, plant family, useful parts of the plant, use categories, and the preparation method for its specific use (if available). Qualitative methods and quantitative ethnobotanical indicators were used for the analysis. To determine the level of awareness of a particular plant or fungus species, absolute frequencies (FC), i.e., the number of interviews in which a particular taxon was mentioned, were first used [74]. Additional use was made of relative frequencies (RFC), i.e., the ratio of FC to the total number of interviews (N = 40) according to Tardio and Pardo-de-Santayana [75].
*RFCs* = *FCs*/*N*,(1)

For each taxon, we then determined the number of use categories, i.e., the number of uses per species (NU), which could range from zero to a maximum of ten available use categories.

For each taxon, a use report (UR), i.e., the total number of uses by all respondents (from *i*_1_ to *i_N_*) and all use categories for that taxon, was calculated. The following formula was used [74]:(2)$${UR}_{S}=\sum_{u=u_{1}}^{u_{NC}}\sum_{i=i_{1}}^{i_{N}}{UR}_{ui}$$

The symbol *NC* stands for the number of different use categories, while *N* stands for the number of interviews or respondents.

The importance of the taxa for the study area was estimated using indices of relative importance (RI) and cultural importance (CI). The index of relative importance takes into account the position of the species in terms of relative abundance and the number of uses according to the taxa that have the highest values of these indicators. The *RI_s_* index is the average of the position of a taxon in relation to the number of surveys in which its use and number of different uses is mentioned:(3)$${RI}_{s}=\frac{{RFC}_{s\left(max\right)}+{RNU}_{S(max)}}{2},$$

In the above formula *RFC_s(max)_* is the ratio between the taxon s and the taxon with the highest *RFC_S_* value, and *RNU_s(max)_* is the ratio between the relative number of uses of *s* and the maximum value [75].

The index of importance for a given taxon s is the ratio of *URs* and the number of respondents *N* [75]:(4)$${CI}_{S}=\sum_{u=u_{1}}^{u_{NC}}\sum_{i=i_{1}}^{i_{N}}{UR}_{ui}/N$$
where *UR_ui_* is the use report for taxon *s*, i.e., the number of uses by all respondents and in all use categories.

The uniformity of the use of plants for the same use categories or the tendency to use a plant for a specific purpose is determined by the fidelity level (FL):(5)$${FL}_{s}=\frac{N_{s}\cdot 100}{{FC}_{s}}$$

The symbol *N_s_* is the number of respondents using plant s for a particular use, and *FC*_s_ is the frequency of mentions for that species [76].

Preliminary data analyses and calculations of descriptive statistics were performed using the program Microsoft Excel Version 2016, Microsoft Corporation, Redmond, WA (USA) [77]. Other calculations and analyses were performed with the package “ethnobotanyR” [78] in R, Version 4.3.2, R Foundation for Statistical Computing, Vienna, Austria [79]. To determine the relationship between the plants in terms of their use, we performed a PCA in PAST 4.03 [80]. For the PCA, we used a matrix with plant names in the rows and use categories in the columns, with the absolute frequencies given as values. The eigenvalues and eigenvectors of the variance–covariance matrix were calculated using the SVD algorithm.

## 4. Conclusions

Ethnobotanical research is a crucial first step for local rural development and small-scale trade in indigenous medicinal and food plants and fungi. High migration rates, depopulation, and aging are typical for the rural continental part of Croatia and lead to an accelerated loss of traditional ethnobotanical knowledge. Our study provides new information on the traditional use of plants in Croatia. New and interesting uses include food uses of *Hosta seiboldiana*, *Pteridium aquilinum*, and *Trapa natans*, and the ethnoveterinary use of *Ambrosia artemisiifolia* and *Sambucus ebulus*. Among the fungi, the use of *Sarcoscypha coccinea* is particularly noteworthy. To summarize, plants and fungi are an integral part of the diet of the local population, and their local names and uses are documented as part of local tradition. We should focus more on documenting this knowledge to facilitate its dissemination in the communities that possess it, or at least to preserve it for future generations.

## Figures and Tables

**Figure 1 plants-13-02153-f001:**
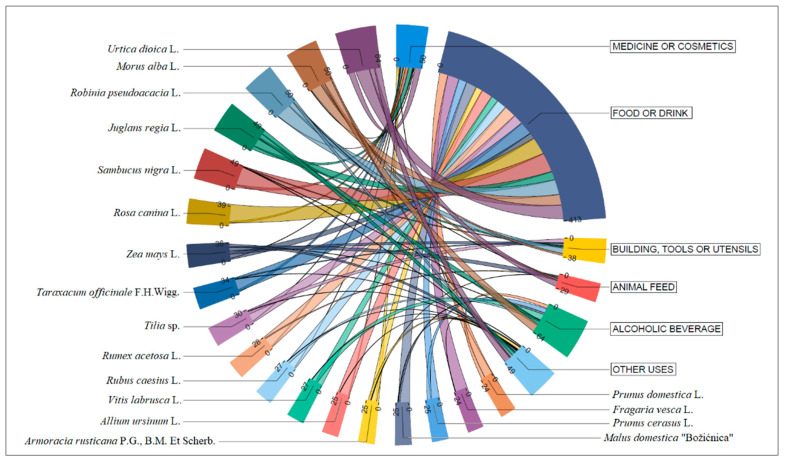
The most used plants in NW Slavonia area according to use categories.

**Figure 2 plants-13-02153-f002:**
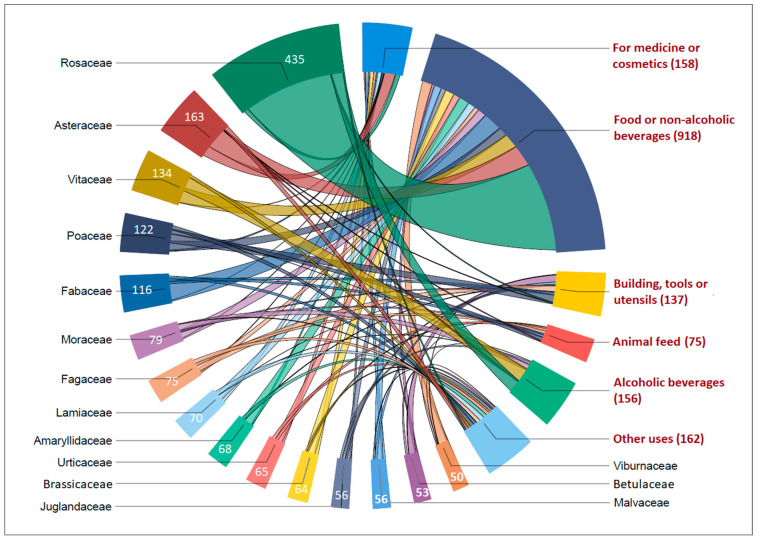
Plant families with the highest use reports in NW Slavonia area by use categories.

**Figure 3 plants-13-02153-f003:**
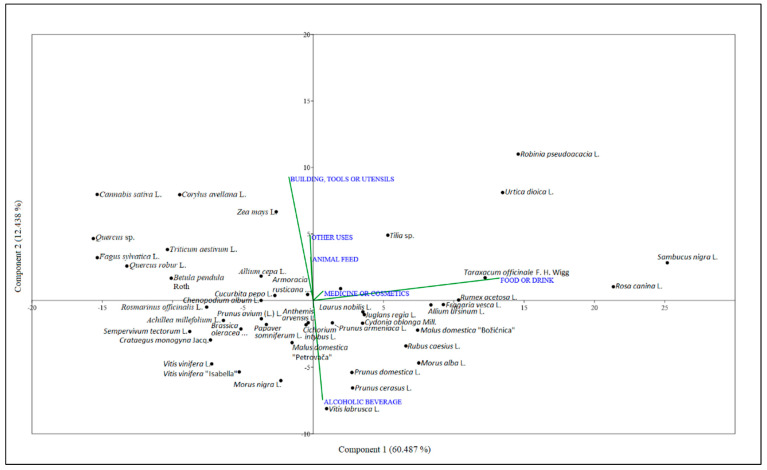
The PCA scatter with biplot for plants with 15 or more use reports.

**Figure 4 plants-13-02153-f004:**
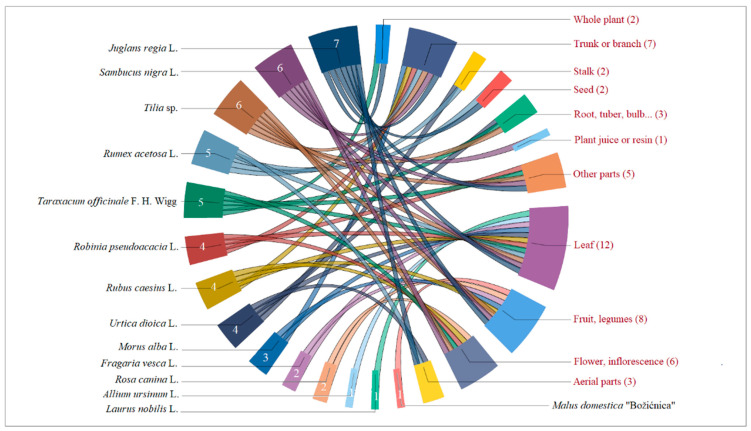
Chord diagram of plants with UR> = 15 and their commonly used plant parts in the NW Slavonia area. *(Note: The numbers next to the plant names indicate the number of plant parts used*).

**Figure 5 plants-13-02153-f005:**
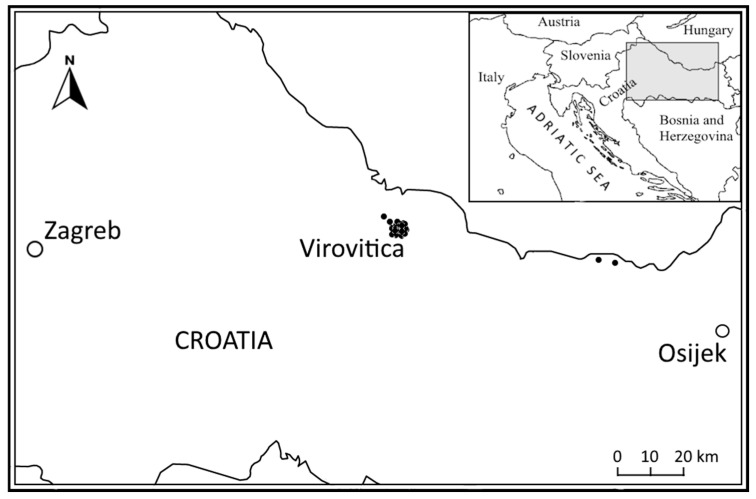
The geographical position of the Northwest Slavonia study site (source: based on the county and municipality maps of Croatia available at https://vemaps.com/croatia/ (accessed on 11 June 2024)).

**Table 1 plants-13-02153-t001:** List of documented wild and cultivated plant taxa used in Northwest Slavonia (Croatia).

Botanical Taxon	Local Name	Frequency	Status	Use Category	Used Part	Use	Written Sources— Regional Examples
*Abies alba* Mill., Pinaceae	jela	4	N	FO, TL, CON	sh, tk	cough syrup; building wood; furniture material	[24]
*Acer campestre* L., Sapindaceae, ZAGR78836	javor	2	N	TL	fr, sd, st	making musical instruments (guitar and “tamburica”) and sleds	
*Achillea millefolium* L., Asteraceae, ZAGR78837	jezičac, purija trava, hajdučica, hiljadarka, stolisnik	10	N	DR, AF, MD, OT	ap, fl, fr, lf, sd, st, wh	tea against menstrual discomfort and stomach problems; purulent wound treatment; tea; for sale; poultry feed; fertilizer	[22,24,25,26,28,30]
*Aesculus hippocastanum* L., Sapindaceae, ZAGR78838	divlji kesten, okrugli kesten	10	C	COS, MD, OT	fr, lf, pj, sd, st	for repelling pests around livestock; macerate against rheumatism and hemorrhoids; poultice against rheumatism; for massage; knee pain reliever (in “rakija”); tincture for varicose veins; stomach pain reliever	[24,25,28]
*Alcea rosea* L., Malvaceae, ZAGR78839	sljez	4	N	FO, DR, MD, OT	fr, lf, sd	against menstrual pain (root steeped in brandy); tea; sold to companies	[22]
*Allium ascalonicum* L., Amaryllidaceae, ZAGR79004	luk kozjak	2	C	FO	fr	cooked as vegetable; against menstrual pain (root in brandy); tea; for sale	[25]
*Allium cepa* L., Amaryllidaceae, ZAGR79005	crveni luk, luk, luk ‘Srebrnjak’, vječno mladi luk	12	C	FO, DR, TL, CE, MD, OT	fr, lf, sd, st	raw; fried or cooked for food; for treating purulent wounds and ulcers; dry scaly leaves for dyeing Easter eggs	[24,25,26,30]
*Allium sativum* L., Amaryllidaceae, ZAGR79006	bijeli luk, češnjak	5	C	FO, TL, CE, MD	fl, fr, rt, sd	raw; fried or cooked as food; against spells (in the past: placed under children’s heads); poultry feed; tincture against hypertension; stomach pain reliever	[25,26]
*Allium schoenoprasum* L., Amaryllidaceae, ZAGR78840	vlasac	5	N / C	FO	ap, fr, lf, rt, sd, st	condiment for salad and stew	[25,26]
*Allium ursinum* L., Amaryllidaceae, ZAGR78841	crijemuš, divlji luk, medvjeđi luk, srijemuš	21	N	FO, MD, OT	ap, bch, bk, fl, fr, in, lf, rt, sd, sh, st, tb, wh	raw salad; mixed with fresh cheese; cooked; blood detox tincture; for sale	[22,25,26,28]
*Alnus glutinosa* (L.) Gaertn., Betulaceae, ZAGR78842	joha	2	N	TL, CON, OT	fr	construction wood for traditional houses; water troughs and garden stakes	
*Aloe barbadensis* (L.) Burm.f., Asphodelaceae	aloa, aloe vera	2	C	FO, COS, MD	fr, lf	eaten raw; medicinal cream; mixed with honeycomb	
*Alopecurus myosuroides* Huds., Poaceae, ZAGR78843	mišji rep, mišji repak	2	N	AF	fr	poultry and other animals’ feed	
*Althaea officinalis* L., *Malvaceae*	bijeli sljez, sljez	2	N / C	DR, MD, OT	fr	tea; drops for stuffy nose; for sale	[22,25]
*Amaranthus retroflexus* L., *Amaranthaceae*, ZAGR78845	štir	3	N	FO	fr, hu, in, lf, sd, sty	leaf and seed for stew	[2,22,25]
*Ambrosia artemisiifolia* L., Asteraceae, ZAGR78846	ambrozija	4	N	MD, OT	fr, in, sd, st, wh	for strengthening blood in horses; came to the region with USA aid after World War II; in the past: for sale	
*Anethum graveolens* L., Apiaceae, ZAGR78847	kopar	11	C	FO	bch, fl, fr, in, lf, sd, tk	sauces; condiment for pickles	[2,26]
*Anthemis arvensis* L., Asteraceae	divlja kamilica, kamilica, samonikla kamilica	16	N	DR, MD	ap, bch, fl, fr, in, lf, sd	tea against cold; inhalation; poultice against pain	[25]
*Anthoxanthum odoratum* L., Poaceae	mirisavka	1	N	MD	fr	poultice against hemorrhoids	
*Arctium lappa* L., Asteraceae, ZAGR79007	čičak	5	N	DR, TL, MD, OT	bch, fr, lf, sd, tk	raw leaf for treating wounds; ulcers and acne; hair growth oil; root tea; children’s toys and playthings (bad if thrown into hair); decoration	[22,24,28]
*Armoracia rusticana* P. Gaertn., B. Mey. et Scherb., Brassicaceae, ZAGR79008	hren	15	N	FO, AF, CE, MD, OT	bch, fr, in, lf, rt, st, tk, tr	raw; freshly grated root for sauce (plenty at Easter); preservative for pickling; leaf tea against headache	[25,28]
*Artemisia absinthium* L., Asteraceae, ZAGR79009	pelin	5	N / C	DR, AL, AF, MD, OT	fr, lf, rt, st, tk	for “pelinkovac” liqueur (good for the stomach; mixed with other species); tea against stomach problems; anthelmintic for animals (leaf; tea); insecticide against aphids	[24,25,26,30]
*Arum maculatum* L., Araceae, ZAGR79010	kozlac	2	N	AF	fr	boiled for pig feed	[24]
*Asarum europaeum* L., Aristolochiaceae	kopitnjak	2	N	OT	hu	sold to the pharmaceutical industry	[31]
*Asparagus densiflorus* (Kunth) Jessop, Asparagaceae, ZAGR79011	asparabus, ukrasna šparoga	1	C	OT	fr	wedding decorations; bouquets	
*Asparagus officinalis* L., Asparagaceae, ZAGR79012	kultivirana šparoga, uzgojena šparoga	2	C		st	soup and risotto; used to be sown (cultivated) extensively	[2,22]
*Avena sativa* L., Poaceae	zob	4	N	FO, AF, TL, CON	ap, fr, lf, rt	food (flour); feed (straw and chaff for cattle; horses); woven baskets and slippers; in the past: mattress filling (“stroža”); roof covering	[24,25,26]
*Balsamita major* Desf., Asteraceae, ZAGR79013	kaloper	4	N / C	MD, OT	fr, lf	kept in vases around the house for the smell; worn behind women’s ears for the smell; tea against laryngitis; tincture and tea against male infertility	
*Bellis perennis* L., Asteraceae, ZAGR79014	tratinčica	3	N	FO, DR, CE, OT	ap, fr	stomach upset tea; flower necklace and decoration; ceremonial face wash on Palm Sunday	[22,24,25,28,30]
*Beta vulgaris* L., Amaranthaceae, ZAGR79015	cikla	3	C	FO, MD, OT	ap, fl, fr, lf	fully cooked and pickled; dyeing fabric and Easter eggs; a piece of beetroot on the heel against fever	[24,25,26]
*Beta vulgaris* L. ssp. *vulgaris* var. *saccharifera* Alef., Amaranthaceae,	šećerna repa	5	C	FO, AF	ap, fr, wh	sugar production; in the past: jam used as sweetener; leaves for food	[25]
*Betula pendula* Roth, Betulaceae, ZAGR79016	breza	9	N	FO, DR, TL, CON, MD, OT	ap, bch, fr, in, lf, sd	bark tea against urological problems; for kidneys; detox; sap rich in minerals collected in early spring; traditional construction material and for furniture; rough brooms	[24,26,28]
*Boswellia sacra* Flueck, Burseraceae	tamjan	2	C	CE	fr, in, sd	protection from evil forces and spells	
*Brassica napus* L. (syn. *Brassica napus* L. ssp. *oleifera* (DC.) Janch.), Brassicaceae, ZAGR79017	uljana repica	2	N	FO	fr, lf	oil production; good bee pastures	[25]
*Brassica napus* ssp. *rapifera* J. Metzg., Brassicaceae	žuta koraba	2	N		hy	raw; cooked in stew	
*Brassica oleracea* L. ssp. *acephala* (DC.) O. Schwarz, Brassicaceae	stočni kelj	2	C	AF	fr	for feeding livestock in winter (cattle; pigs; poultry)	[25]
*Brassica oleracea* ssp. *capitata* (L.) Duchesne, Brassicaceae	kupus, kupus ‘Varaždinac’, zelje	12	C	FO, MD	ap, fr, in, lf, sd, st, tk	raw; cooked; base for baking bread; pickled for stuffed leaves with minced meat (sarma); poultice from the leaves used against fever or mastitis; reduces inflammation; helps with sprains; oiled leaves placed on burns	[24,25,26]
*Brassica rapa* L., Brassicaceae	bijela repa, crvena repa, repa, žuta repa	10	N/C	FO, AF	fr, lf, sd, st, tk	cooked food; pickled; filling for strudel; feed	[25,26]
*Briza media* L., Poaceae	majčine suzice	2	N	DR, OT	ap, sd	tea against diarrhea; decoration	
*Buxus sempervirens* L., Buxaceae, ZAGR79018	bušpan	2	C	CE, OT	sd, tk	decorative fence or hedge; for graves for All Saints Day	[31]
*Calendula officinalis* L., Asteraceae, ZAGR79019	neven	5	N / C	DR, COS, MD, OT	ap, bch, fr, lf, tk	petals used in medicinal ointment for skin (acne) and varicose veins; against moles; tea against hypertension (with nettle leaf); tincture against heart problems	[22,24,25,26,30,32]
*Cannabis sativa* L., Cannabaceae, ZAGR79218	konoplja, kudelja konoplja	16	C	FO, TL, CON, OT	fl, fr, in, lf, sd, st, tk, wh	for ropes; linen; sacks and blankets; filling for traditional mattress “strože”, seed oil; for sale	[22,24,25,26]
*Capsella bursa-pastoris* (L.) Medik., Brassicaceae, ZAGR79020	rusomača	2	N	MD, OT	fl, fr, pj	tea against urological and menstrual problems; the fruit is balanced on the thumb as a game	[28]
*Capsicum annuum* L., Solanaceae, ZAGR79117	paprika ljuta, feferona	4	C	FO, TL	ap, fl, in, sd	raw; cooked; condiment	[25,26]
*Capsicum annuum* ‘Babura’, Solanaceae, ZAGR79156	paprika ‘Babura’	2	C	FO	fl, in	raw; pickled	
*Capsicum annuum* ‘Paradajzerica’, Solanaceae, ZAGR79216	paprika ‘Paradajzerica’	2	C	FO	fl, fr	raw; pickled	
*Capsicum annuum* ‘Rog’, Solanaceae, ZAGR79217	paprika ‘Rog’	3	C	FO	fl, fr, lf	raw; “ajvar” ingredient	
*Carpinus betulus* L., Betulaceae, ZAGR79021	grab, grab majpan	9	N	AF, TL, CE, OT	fl, fr, lf, pj	for the celebration of May 1st (“majpan”); firewood; tool handles; litter; sawdust for smoking; toys	[31,33]
*Carum carvi* L., Apiaceae	kim	4	N / C	FO, MD	fr, lf, pj, rh, wh	condiment; against flatulence	[22,24,25,26]
*Castanea sativa* Mill., Fagaceae, ZAGR79022	kesten, kresten, pitomi kesten	18	N / C	FO, TL, CON, OT	bch, fr, hy, in, lf, sd, st, tb, tk	boiled or baked; flour; leaves as a baking mat; poles; building wood; for sale	[7,22,25,26]
*Centaurium erythraea* Rafn, Gentianaceae	kičica	1	N	MD	lf	tea against stomach problems	[22,28]
*Chamomilla recutita* (L.) Rauschert, Asteraceae, ZAGR79023	kamilica	4	C	DR, MD	bk, fl, fr, lf, pj, sd	calming tea; tea and macerate for stomach; health and immunity; poultice for cleaning eyes; hair wash	[22,24,25,26,28,30]
*Chelidonium majus* L., Papaveraceae, ZAGR79024	cimbola, rosopas	5	N	MD, OT	bd, fl, hu, lf, st	cell sap against warts; for sale	[22,25,28]
*Chenopodium album* L., Amaranthaceae, ZAGR79025	loboda	16	N / C	FO, AF, OT	bch, fr, hy, in, lf, rt, sd, st	cooked (stew; strudel) or prepared as spinach; feed (with potato peel); insecticide against aphids and potato beetle; in the past: sown	[2,22,25]
*Chrysanthemum* x *morifolium* (Ramat.) Hemsl., Asteraceae, ZAGR78923	krizantema	2	C	CE	fr	graveyard decoration (All Saints Day)	[30]
*Cichorium endivia* L., Asteraceae	andivija, endivija	2	C	FO	fr, lf	raw salad	
*Cichorium intybus* L., Asteraceae, ZAGR79026	cikorija, vodopija	11	N / C	FO, DR, MD	bl, fl, fr, in, st, tb	raw (young leaves) with pork fat; or cooked (stew with potato); coffee substitute	[2,22,28]
*Colchicum autumnale* L., Colchicaceae, ZAGR79027	mrazovac	14	N	MD, OT	bch, fl, fr, in, lf, sd	dried for sale; or decoration in pots	[25]
*Cornus mas* L., Cornaceae	drenak, drijen	3	N	FO, DR, AL, CE, OT	fl, lf, tb, tk	for distillation (“rakija”) or brandy flavoring; liqueurs; tea; jam; flower branch for blessing on Palm Sunday in the church	[22,24,25,26]
*Cornus sanguinea* L., Cornaceae, ZAGR79028	svib	1	N	TL	fr, lf	slingshot making (children)	
*Corylus avellana* L., Betulaceae, ZAGR79029	divlji lješnjak, lijeska, lješnjak, šumski lješnjak	14	N / C	FO, TL, CON, OT	ap, bch, fl, fr, lf, rt, sd, st, wh	raw fruit; cake; building of traditional houses; farming utensils (three-horn hay fork); fishing rod; stick; handles; garden stalks; wicker fence (with ivy); branches for woven baskets; bows and arrows for children; whistles; in the past: twigs for punishing children; for sale	[24,25,26]
*Crataegus monogyna* Jacq., Rosaceae, ZAGR79030	bijeli glog, glog	10	N	FO, DR, AL, TL, MD	ap, bch, fl, fr, hy, lf, rt, st, wh	raw; tea against hypertension and insomnia; tincture for the treatment of heart disease; for distillation (“rakija”); liqueurs; slingshot	[24,25,26,28]
*Crataegus nigra* Waldst. et Kit., Rosaceae	crni glog, glog	2	N	MD, OT	ap, sd	tea against hypertension	
*Cucurbita pepo* L., Cucurbitaceae, ZAGR79119	buča, bundeva, mesirka, misirka, turkinja (bundeva), valjanka	15	C			baked or in stew; especially on Christmas Eve and in periods of fasting; roasted seeds for snacking; a highly valued oil; feed (“pogača”) for cattle and pigs; custom: expressing love by carving a name on the fruit	[26]
*Cucurbita pepo* L. ‘Mađerka’, Cucurbitaceae	mesirka ‘Mađerka’	1	C	FO	lf	soup or baked	
*Cucurbita pepo* L. ‘Turkinja’, Cucurbitaceae	bundeva ‘Turkinja’, mesirka ‘Turkinja’	2	C	FO, AF	ap	tsew; oven baked; feed (pig and cattle); roasted seeds for snacking	
*Cyclamen purpurascens* Mill., Primulaceae	košutica, šumska ciklama	2	N / C	OT	bl, fr	decoration in house and garden	[30]
*Cydonia oblonga* Mill., Rosaceae, ZAGR79031	dunja	17	C	FO, DR, AL, MD, OT	ap, bl, fl, fr, in, lf, sd, st, tb, wh	compote; jam; quince cheese or paste (“kitnikez”); for distillation (“brandy”); liqueurs; anti-diarrhea and anti-dysentery tea leaf; kept in wardrobes throughout the winter for fragrance	[24,25,26]
*Daucus carota* L. ssp. *sativus* (Hoffm.) Arcang., Apiaceae, ZAGR79032	mrkva	1	C	FO	st	raw; cooked; condiment; was sown	[25,26]
*Dryopteris filix-mas* (L.) Schott, Dryopteridaceae, ZAGR79033	paprat muška, paprat	4	N	MD, OT	fr, in, lf, pj, st	agent against red mites; ticks; fleas and lice; honeybee protection against varroa; decoration; for sale	[31,34]
*Elymus repens* (L.) Gould, Poaceae, ZAGR79034	pirika	5	N	DR, MD	ap, fr, pj, rt	pain reliever tea; against lung problems; fertilizer; for sale	[22,24]
*Equisetum arvense* L., Equisetaceae, ZAGR79035	poljska preslica, preslica	8	N	FO, DR, MD, OT	bk, fl, fr, in, lf, lg, rt, sd	tea against urological problems; for prostate; against feet sweating; insecticide against aphids; fertilizer; for sale	[22,25,28]
*Fagopyrum esculentum* Moench, Polygonaceae	heljda	2	N / C	FO	in, rh	side dish; flour	[25,26]
*Fagus sylvatica* L., Fagaceae, ZAGR79037	bukva	11	N	AF, TL, CON, MD, OT	bch, fr, in, lf, pj, st, tk, wh	leaf to stop bleeding; leaf for treating ulcers and warts; pig feed; litter for livestock; building wood; for tools; toys and utensils; furniture materials; firewood; sawdust for smoking meat; game: “masličanje” in which a pointed beech pole was to be thrown so that it would land upright in the ground, and the opponent had to knock it over with another pole	[22,24,26]
*Festuca pratensis* Huds., Poaceae, ZAGR79038	trava, vlasulja	2	N	TL, OT	fr	whistling on leaves	
*Ficus carica* L., Moraceae, ZAGR79039	smokva	2	C	FO, AL, MD	fl, lf	raw; jam; boiled dry figs for digestion	[30]
*Forsythia* sp., Oleaceae, ZAGR79040	forzicija	3	C	CE	fl, lf, rt	Palm Sunday ceremony	
*Fragaria vesca* L., Rosaceae, ZAGR79041	divlja jagoda, jagoda, šumska jagoda	21	N	FO, DR, OT	ap, bch, fl, fr, in, lf, lg, rt, sd	raw; jam; dried leaves for tea; in gift bouquets	[22,24,25,26,28]
*Frangula alnus* Mill., Rhamnaceae	krkovina, tušljika	2	N	MD, OT	ap, bch, fr	for sale (pharmaceutical and medical purposes)	[35]
*Fraxinus angustifolia* Vahl, Oleaceae	jasen	2	N	AF, TL	fr, in	animal feed; furniture making	[36]
*Galanthus nivalis* L., Amaryllidaceae, ZAGR79042	visibaba	2	N	TL, OT	bl, lf	decoration in the house and garden	[24]
*Galium odoratum* (L.) Scop., Rubiaceae, ZAGR79210	lazarkinja	2	N	DR, AL, OT	lf	tea; flavoring for brandy (“rakija lazarkinja”) and cigarettes	[37]
*Glechoma hederacea* L., Lamiaceae, ZAGR79043	dobričica	2	N	MD, OT	lf	pain reliever ointment (cooked in pig fat); for sale	[37]
*Hedera helix* L., Araliaceae, ZAGR79044	bršljan	6	N / C	FO, TL, CON, CE, MD	fr, lf	cough syrup and tea; wicker fence combined with *Corylus avellana*; funeral and Easter decorations	[24,26]
*Helianthus annuus* L., Asteraceae, ZAGR79120	suncokret	2	C	FO	bch, fl, fr	seeds for snacking; oil; good as bee pasture (in Baranja)	[25]
*Helianthus tuberosus* L., Asteraceae, ZAGR79045	čičoka, divlji krumpir	5	N / C	FO, DR, AL, AF, MD	ap, fr, in	raw; food for diabetics; feed; horse feed for speed; for distillation (it was poisonous “rakija” to deceive alcoholics); liqueurs	[22,25]
*Helichrysum italicum* (Roth) G. Don, Asteraceae	smilje	2	C	COS, MD, OT	ap	ointment; garden decoration	[22,30]
*Herniaria hirsuta* L., Caryophyllaceae	kilavica	2	N	OT	wh	for sale (pharmaceutical and medical purposes)	
*Hordeum vulgare* L., Poaceae	ječam	6	N	FO, DR, AF, CE	fl, fr, hu, in, sd, tk	bread “for the poor”, side dish; barley porridge “geršl”, coffee substitute; straw for animal feed; straw mattress protects against groundwater	[24,25,26]
*Hosta sieboldiana* (Hook.) Engl., Asparagaceae, ZAGR79046	hosta	2	C	FO	ap, fr	for “sarma” (minced meat with rice wrapped in a leaf); decoration in garden	
*Humulus lupulus* L., Cannabaceae, ZAGR79047	divlji hmelj, hmelj	5	N	FO, DR, AL	fl, fr, lf, wh	dried for tea; brewing beer; young shoots fried with eggs; in the past: the flower was eaten	[22]
*Hyoscyamus niger* L., Solanaceae	crna bunika	2	N	OT	fr	boiled fruit (jam) as poison; in the past: hallucinogenic agent	[31,38,39]
*Hypericum perforatum* L., Hypericaceae, ZAGR78844	gospina trava, kantarion	3	N	DR, MD	ap, fr, in, lf	tea against depression and for regulating menstrual problems; strengthening immunity; red medicinal oil for the treatment of skin diseases (psoriasis and burns)	[24,25,26,28,30]
*Juglans nigra* L., Juglandaceae	crni orah	3	C	FO, TL	ap, fr, sd	rifle stock	Not in the region but used for this in N. America [40,41]
*Juglans regia* ‘Koštunac’, Juglandaceae, ZAGR79048	orah ‘Koštunac’	2	C	FO	fr, lf	raw food for diabetics; cake “orahnjača”, brandy “orahovac” for the stomach; liqueur for treating thyroid problems; dyeing fabrics; furniture materials; garden stakes; stock; leaf: insect and moth repellent (put in *Phaseolus vulgaris* with just few grains *Piper nigrum*, it was to expensive); smoked like tobacco; for slingshot and helicopter games	
*Juglans regia* L., Juglandaceae	crveni orah, orah	24	C	FO, AL, TL, MD, OT	ap, bl, fr, in, lf, rh, sd, st	for “orahnjača” cake	[24,25,26,28,30]
*Juncus effusus* L., Juncaceae, ZAGR79049	zukva trava	1	N	TL	fr, lf	against throat pain	
*Juniperus communis* L., Cupressaceae	borovica	3	N	FO, AL, TL, MD	lf, lg	in brandy against diarrhea; against throat pain; condiment for game meat	[22,24]
*Lactuca sativa* ‘Hrastov list’, Asteraceae	salata ‘Hrastov list’	2	C	FO	fr, in, sd	raw salad	[25,26]
*Lagenaria siceraria* (Molina) Standl., Cucurbitaceae, ZAGR79121	tikvica, ukrasna tikvica, ukrasna tikvica ‘zug’	5	C	TL	bch, fr, lf	ladle for taking wine out of a barrel; swimming aid; decoration; toy	
*Lamium maculatum* (L.) L., Lamiaceae, ZAGR79050	pjegava kopriva	2	N	FO	bch, in	nectar from flowers as a raw children’s snack	[33]
*Laurus nobilis* L., Lauraceae, ZAGR79051	lovor	20	N / C	FO, DR, MD, OT	ap, fl, fr, in, lf, sd, st, wh	cough syrup; laurel tea drunk for 14 days against bronchitis; condiment in cooked food and sauerkraut	[22,30]
*Lavandula intermedia* Emeric ex Loisel., Lamiaceae, ZAGR79061	lavanda	12	C	COS, OT	bch, fl, fr, in, lf, rt, sd, st, tk, wh	dried against moths and flies; relaxing agent for better sleep; in massage oil	[26]
*Levisticum officinale* W. D. J. Koch, Apiaceae, ZAGR79122	ljupčac, ‘magi’, ‘vegeta’ biljka	4	C	FO	fl, fr, in, wh	condiment	[26]
*Lilium candidum* L., Liliaceae	antunovski ljiljan, bijeli ljiljan sv Ante, bijelo blaženo, ljiljan	9	C	CE, MD	fr, in, pj, sd	ointment against skin burns; bouquets of lilies brought to church on St. Anthony’s Day (June 13) and for the rite of baptism	[25,30]
*Linum usitatissimum* L., Linaceae	lan	9	N / C	FO, AF, TL, OT	ap, fr, in, lf, sd	food; feed (seed and bran digestive agent for cattle); fibers for linen cloth; used to be woven; was cultivated but not extensively	[22,24,25]
*Malus domestica* (Suckow) Borkh., Rosaceae	jabuka	7	C	FO, DR, AL, TL	fl, fr, lf, st	raw; dried; compote; juice; vinegar; for distillation (“rakija”); roasted on open fire on a stick; stored in cereal grains (or underground storage; a so-called “trap”)	[22,24,25,26]
*Malus domestica* ‘Božićnica’, Rosaceae	jabuka ‘Božićnica’, jabuka ‘Koturača’, jabuka ‘Pogačarka’, jabuka ‘Tanjurača’	21	C	FO, DR, AL, CE	bch, fl, fr, hy, in, lf, sd, st, wh	raw; juice; compote; dried slices; oven baked; made into vinegar; for distillation “rakija”); blessed on Candlemas	[26]
*Malus domestica* ‘Crveni delišes’, Rosaceae	jabuka ‘Crveni delišes’	1	C	FO	fr	dried	
*Malus domestica* ‘Debeljara’, Rosaceae	jabuka ‘Debeljara’	2	C	FO	fr, tk	raw; cakes; juice	
*Malus domestica* ‘Idared’, Rosaceae	jabuka ‘Idared’	2	C	FO	fr, lf, st, tk	raw; dried	
*Malus domestica* ‘Ivančica’, Rosaceae	jabuka ‘Ivančica’	1	C	FO	lf, st	raw	
*Malus domestica* ‘Jonagold’, Rosaceae	jabuka ‘Jonagold’	2	C	FO, AL	lf, rt, st	raw; dried; for distillation (“rakija”)	
*Malus domestica* ‘Jonator’, Rosaceae	jabuka ‘Jonator’	1	C	FO	lg	dried	
*Malus domestica* ‘Kanada’, Rosaceae	jabuka ‘Kanada’, jabuka ‘Musavka’	2	C	FO	ap	raw; dried; compote; cake	[26]
*Malus domestica* ‘Katarinčica’, Rosaceae	jabuka ‘Katarinčica’	1	C	FO	bch	raw	
*Malus domestica* ‘Kožara’, Rosaceae	jabuka ‘Kožara’, jabuka ‘Kožnjara’	2	C	FO	fl	raw; compote	
*Malus domestica* ‘Limonka’, Rosaceae	jabuka ‘Limunka’	2	C	FO	fr, lf, sd	apple vinegar; cake; oven baked	
*Malus domestica* ‘Mašanka’, Rosaceae	jabuka ‘Mašanka’	2	C	FO	fr, hy	raw; juice; cake	
*Malus domestica* ‘Musavka’, Rosaceae	jabuka ‘Musavka’	5	C	FO	bch, fr, im, lf, sd, tb	raw; dried; kept in “trap” (traditional underground storage)	
*Malus domestica* ‘Paradija’, Rosaceae	jabuka ‘Paradija’	2	C	FO	fr	raw; apple vinegar; cake; oven baked	
*Malus domestica* ‘Petrovača’, Rosaceae, ZAGR79211	jabuka ‘Petrovača’, jabuka ‘Petrovka’	12	C	FO, DR, AL	bch, fr, hy, lf, sd, wh	raw; grated and dried; juice; compote; cake; for distillation (“rakija”); vinegar	[26]
*Malus domestica* ‘Slavonska srčika’, Rosaceae	jabuka ‘Srčika’	1	C	FO	fl	raw; kept in “trap” (traditional underground storage)	
*Malus domestica* ‘Starking’, Rosaceae	jabuka ‘Starking’	1	C	FO	hy	raw	
*Malus domestica* ‘Zelenika’, Rosaceae	jabuka ‘Zelenika’, jabuka ‘Zelenjavka’	3	C	FO	fr, lf	raw; vinegar; grated and dried; for cakes	
*Malus domestica* ‘Zlatni delišes’, Rosaceae	jabuka ‘Zlatni delišes’	1	C	FO	ap	dried	
*Malus sylvestris* (L.) Mill., Rosaceae, ZAGR79062	divlja jabučica, divlja jabuka, jabučica divlja	7	N	FO, AL	fr, lf, sd, st	raw; dried; compote; in jam mixed with other fruits; for distillation (“rakija”)	[22,25]
*Malva sylvestris* L., Malvaceae, ZAGR79063	crni sljez, sljez	2	N	DR, MD	bk, fr, in, lf	cough tea; expectorant; root decoction for rinsing the nose	[28]
*Medicago sativa* L., Fabaceae	lucerna	3	N	AF	fr, sd, sh	animal feed	
*Melissa officinalis* L., Lamiaceae, ZAGR79064	matičnjak, melissa	7	C	FO, DR, MD, OT	fl, fr, in, lg, sh	calming tea; against headache; against menstrual pain; fresh salad; syrup; condiment	[22,24,25,28]
*Mentha arvensis* L., Lamiaceae, ZAGR79065	divlja menta, metvica	2	N	DR, MD	fl, lf, st	tea against cough	[24,28,30]
*Mentha* x *piperita* L., Lamiaceae	menta, pitoma menta	4	C	FO, DR, MD, OT	fr, lf, pj, sd	tea against cramps and indigestion; stomach problems; poultice against rheumatism and toothache; mosquito repellent	[24,25,28,30]
*Mespilus germanica* L., Rosaceae, ZAGR79066	divlja mušmula, mušmula, mušmulja	9	N / C	FO	fr, in, lf, rt	raw; jam; kept in hay to fully mature	[24,25]
*Morus alba* L., Moraceae, ZAGR79067	bijeli dud, žuti dud	26	C	FO, DR, AL, AF, TL, MD, OT	ap, bch, fr, in, lf, rt, st, tk, tr	raw; juice; jam; for distillation (“rakija dudovača”); poultry feed; leaf tea lowers blood sugar; barrels for “rakija”, in the past: for mulberry silkworm	[11,12,22,25]
*Morus nigra* L., Moraceae	crni dud, crveni dud	12	C	FO, DR, AL, AF, TL	ap, bch, fr, lf, sd, st, tk	raw; juice; jam; for spirit distillation (“rakija dudovača”); poultry feed; barrels for “rakija”	[11,12,22,25]
*Narcissus pseudonarcissus* L., Amaryllidaceae, ZAGR79068	zelenkada	2	C	CE	fr, lf	Palm Sunday ceremony; bouquets as decoration	[26]
*Nasturtium officinale* R. Br., Brassicaceae	potočarka	2	N	FO	lf	fresh salad	[2,22]
*Nicotiana tabacum* L., Solanaceae	duhan, duhan ‘Berlej’, duhan ‘Berlejac’	9	N / C	OT	fl, fr, lf, rt, sd	smoking leaves; fly; moth; aphid repellent	[24,26]
*Nymphaea alba* L., Nymphaeaceae, ZAGR79123	lopoč	1	N	TL	bch	decoration	
*Ocimum basilicum* L., Lamiaceae, ZAGR79069	bosiljak	2	N	FO	lf	condiment	[30]
*Olea europaea* L., Oleaceae	maslina	2	C	FO, CE	fr	healthy oil; blessed on Palm Sunday	[22]
*Origanum vulgare* L., Lamiaceae	divlji mažuran, origano	2	N / C	FO, DR, MD	lf, fl	condiment; tea	[22,28,30]
*Oxalis acetosella* L., Oxalidaceae	kiseli cecelj	1	N		lf	young leaves as raw snack	
*Papaver rhoeas* L., Papaveraceae, ZAGR61051	divlji mak	3	N	FO, MD	fr, lf, sd	calming tea; capsules (“škropulja”) were sold to the food industry	[2,22,25,30]
*Papaver somniferum* L., Papaveraceae	mak	10	C	FO, DR, AF, MD	ap, fl, fr, lf, sd	for cakes: “makovnjača”, calming tea for children and animals; seeds in “rakija” against insomnia; decoration	[25]
*Passiflora caerulea* L., Passifloraceae	kristov trn	1	C	OT	ap	decoration	
*Pelargonium zonale* (L.) l’Hér, Geraniaceae, ZAGR79070	muškatl, pelargonija	2	C	COS, OT	lf	hair decoration on festive occasions (for girls ready for marriage); garden decoration	[26,30]
*Petroselinum crispum* (Mill.) A. W. Hill, Apiaceae, ZAGR79071	peršin	2	C	FO, MD	fr, lf	condiment; tea against bacterial poisoning	[26,30]
*Phacelia tanacetifolia* Benth, Boraginaceae	facelija	2	C	FO, OT	lg	honeybee pasture; green fertilizer	
*Phaseolus vulgaris* L., Fabaceae, ZAGR79124	grah	3	C	FO, OT	fr, st	cooked; for children’s mill game	[24,25]
*Phaseolus vulgaris* ‘Mahunar’, Fabaceae	grah ‘Mahunar’	2ut paper	C	FO	fr	cooked for salad and in stew	[26]
*Phaseolus vulgaris* ‘Božićni’, Fabaceae	grah ‘Božićni’	1	C	FO	bch	cooked	
*Phaseolus vulgaris* ‘Žuta olovka’, Fabaceae, ZAGR79212	mahuna ‘Žuta olovka’, visoke mahune	8	C	FO	fl, fr, lf, rt, tk	cooked in stew	
*Phaseolus vulgaris* ‘Prdov’, Fabaceae	grah ‘Prdov’	3	C	FO	ap, fr, sd, st	cooked in stew	
*Phaseolus vulgaris* ‘Puterfizol’, Fabaceae	grah ‘Puterfizol’	3	C	FO	fr, lf, sd	cooked for salad and in stew	
*Phaseolus vulgaris* ‘Tetovac’, Fabaceae	grah ‘Tetovac’	2	C	FO	lf, tk	cooked in stew	
*Phaseolus vulgaris* ‘Trešnjevac’, Fabaceae	grah trešnjar, grah ‘Trešnjevac’	7	C	FO	fr, in, lf, sd	cooked for salad; soup; in stew; side dish	
*Phaseolus vulgaris* ‘Zelenček’, Fabaceae	grah ‘kukuružnjak’, grah ‘Zelenček’, grah ‘Zelenčok’, grah ‘Zelenščak’	7	C	FO, OT	fl, fr, lf, sd, st, tb, tk, wh	cooked for salad; in stew; children’s games with ear leaves and bean: mill game (local: “školice”)	
*Phragmites australis* (Cav.) Steud., Poaceae, ZAGR79072	trska, trstika	7	N	TL, CON	bk, fl, fr, lf	young shoots for snacking; cooked root; for flour; building material in traditional houses; shelter from the sun	
*Picea abies* (L.) H. Karst., Pinaceae	smreka	3	N	TL, CON, MD	sh, lf, tr	cough syrup; raw against lung problems; building wood; garden stake	[24,26,28]
*Pinus nigra* J. F. Arnold, Pinaceae	bor, bor crni	3	C	CE, MD, OT	lf, sh, wh, rs	pine needle syrup against cough and bronchitis; glue resin; Christmas tree decoration	[33]
*Pinus sylvestris* L., Pinaceae, ZAGR79155	šumski bor	2	C	MD	lf, sh	pine needle syrup against cough and bronchitis	[26,28]
*Pisum sativum* L., Fabaceae, ZAGR79125	grašak ‘Mahunar’, grašak ‘Telefon’, ljubičasti grašak	3	C	FO	fr	cooked side dish	[25,26]
*Plantago lanceolata* L., Plantaginaceae, ZAGR79073	bokvica, traputac, trputac, uskolisna bokvica, uskolisni trputac	4	N	MD	lf	tea and syrup against cough; fresh leaf for treating wounds and ulcers	[24,25,28]
*Plantago major* L., Plantaginaceae, ZAGR79074	bokvica, širokolisni trputac, traputac, trputac, veliki trputac	12	N	MD	lf	tea and syrup against cough; fresh leaf for treating wounds and burns; for faster healing	[24,25,26,28]
*Populus alba* L., Salicaceae	topola	2	N	TL, CON	ap, tr	construction wood in traditional houses (floors and ceilings; slats; easels); furniture material	[36,42]
*Portulaca oleracea* L., Portulacaceae, ZAGR79075	tušt	6	N	FO, DR, AF	lf, ap	fresh salad; pig feed	[2]
*Primula vulgaris* Huds., Primulaceae, ZAGR79076	jaglac	2	N	FO, MD, OT	lf, fl	fresh flowers and leaves for salad; tea and syrup against cough; tea against constipation; insomnia; menstrual discomforts	[24,28]
*Prunus armeniaca* L., Rosaceae, ZAGR79126	kajsija, marelica	16	C	FO	fr, sd	raw; compote; jam; peach seeds eaten like almonds; grinded seed in cakes	[25,26]
*Prunus armeniaca* ‘Mađarska najbolja’, Rosaceae	kajsija ‘Mađarica’	2	C	FO	fr, sd	raw; jam	
*Prunus avium* (L.) L., Rosaceae, ZAGR79077	divlja trešnja, trešnja ‘Cepika’	13	C	FO, AL, TL, CON, MD	fr, tr, ap, rs, bch	raw; tea from twigs for better blood flow; for distillation (“rakija”); furniture material; resin as glue	[22,24,25,26]
*Prunus avium* ‘Hrušt’, Rosaceae	trešnja ‘Hrušt’, trešnja hruštavka	5	C	FO, MD	fr, sd	raw; compote; endocarp in pillows against rheumatism	
*Prunus avium* ‘Spasovka’, Rosaceae	trešnja ‘Spasovka’	2	C	FO	fr	raw	
*Prunus cerasifera* Ehrh., Rosaceae, ZAGR79078	ringlov	3	C	FO, AL	fr	raw; for distillation (“rakija”)	[11]
*Prunus cerasus* L., Rosaceae, ZAGR79079	višnja	12	N	FO, DR, AL	ap, bch, fl, fr, lf, st, tk	raw; juice; jam; compote; syrup; tea from a young twig; for distillation (“rakija”); “višnjevac” liqueur	[25,26]
*Prunus cerasus* ‘Španjolka’, Rosaceae	višnja ‘Španjolka’	2	C	FO, DR, MD	fr, bch, sd	raw; compote; tea from young twigs; endocarp in pillows against rheumatism	
*Prunus domestica* L., Rosaceae, ZAGR79128	šljiva, šljiva plava	14	C	FO, DR, COS, AL	fr	raw; jam; compote; dried; for distillation (“rakija”); coloring of brandy	[24,25,26]
*Prunus domestica* ‘Bistrica’, Rosaceae	šljiva ‘Bistrica’	7	C	FO, AL	fr	raw; dried; compote; jam; for distillation (“rakija”)	[26]
*Prunus domestica* ‘Čačanka’, Rosaceae	šljiva ‘Čačanka’	3	C	FO, AL	fr	raw; compote; for distillation (“rakija”)	
*Prunus domestica* ‘Debeljara’, Rosaceae	šljiva ‘Debeljara’	3	C	FO	fr	raw; compote	
*Prunus domestica* ‘Jajara’, Rosaceae	šljiva ‘Jajara’	2	C	FO	fr	raw; jam	
*Prunus domestica* ‘Kalanka’, Rosaceae	šljiva ‘Kalanka’	3	C	FO	fr	raw; dried; compote	
*Prunus domestica* ‘Požežanka’, Rosaceae	šljiva ‘Požežanka’	3	C	FO, AL	ap, fr	raw; dried; jam; for distillation (“rakija”)	
*Prunus domestica* ‘Stenlijevka’, Rosaceae	šljiva ‘Stenlijevka’	3	C	FO	ap, fr, lf	raw; jam	
*Prunus persica* (L.) Batsch, Rosaceae, ZAGR79127	breskva	2	C	FO	bch, tb	raw; jam; compote	[26,30]
*Prunus persica* ‘Golica’, Rosaceae	breskva ‘Golica’	1	C	FO	fr	raw	
*Prunus persica* ‘Vinogradarska’, Rosaceae	breska ‘Vinogradarska’	4	C	FO, AL	fr, lf, sd, st	raw; jam; for distillation (“rakija”)	
*Prunus spinosa* L., Rosaceae, ZAGR79151	divlja šljiva, trnina	8	N	FO, AL, TL	fr, ap	raw; dried for tea; liqueur; juice; syrup; jam; trunk hayfork	[22,24,28]
*Pteridium aquilinum* (L.) Kuhn, Dennstaedtiaceae, ZAGR79080	bujad	1	N	FO	sh	young shoots prepared as cooked salad like asparagus	[22]
*Pulmonaria officinalis* L., Boraginaceae, ZAGR79081	plućnjak	5	N	FO, DR, MD, OT	fl, lf	tea for respiratory system; lungs; against cough; fresh leaves eaten; for sale	[24,25,28]
*Pyrus communis* L., Rosaceae, ZAGR79129	kruška	5	C	FO, AL	fr, sd	raw; dried; compote; for distillation (“rakija”)	[24,25,26]
*Pyrus communis* ‘Citronka’, Rosaceae	kruška ‘Citronka’, kruška ‘Limonka’	4	C	FO	fr	raw	
*Pyrus communis* ‘Gelertova’, Rosaceae	kruška ‘Gelertova’	1	C	FO	fr	raw	
*Pyrus communis* ‘Jagodarka’, Rosaceae	kruška ‘Jagodarka’	1	C	FO	fr	raw	
*Pyrus communis* ‘Lubeničarka’, Rosaceae	kruška ‘Lubeničarka’	2	C	FO	fr	raw; dried	
*Pyrus communis* ‘Petrovka’, Rosaceae	kruška ‘Petrovka’	3	C	FO	fr	raw	
*Pyrus communis* ‘Santa Maria’, Rosaceae	kruška ‘Santa Maria’	1	C	FO	fr	raw	
*Pyrus communis* ‘Tepka’, Rosaceae	kruška ‘Tepka’	2	C	FO, AL	fr	raw; for distillation (“rakija”)	[24,26]
*Pyrus communis* ‘Tikvica’, Rosaceae	kruška ‘Tikvica’	7	C	FO, DR, AL	fr	raw; dried; compote; liqueur “kruškovac”	
*Pyrus communis* ‘Vilijamovka’, Rosaceae	kruška ‘Vilijamovka’	2	C	FO, AL	fl, fr	raw; for distillation (“rakija”); fruit is grown in a glass bottle	
*Pyrus communis* ‘Žetvenjača’, Rosaceae	kruška ‘Žetvenjača’	2	C	FO	fr	raw	
*Pyrus pyraster* (L.) Burgsd., Rosaceae	divlja kruška	2	N	FO	fr	raw; dried	[22,24,25]
*Quercus petraea* (Matt.) Liebl., Fagaceae, ZAGR79152	hrast kitnjak	3	N / C	AF, TL, CON, OT	fr, lf, tr, bch	pig feed; garden stakes; whistles from acorn caps; collected for sale to forest authorities for afforestation	[24,25,26]
*Quercus robur* L., Fagaceae, ZAGR79153	hrast lužnjak	8	N / C	FO, AF, TL, CON, OT	ap, fr, lf, tk, bch, bk	pig feed; garden stakes; acorns sold to state forestry for afforestation; wood for making barrels; building wooden furniture and houses; tanning leather with oak bark	[24]
*Quercus* sp. (include *Q. petraea, Q. robur* and *Q. cerris* L.) Fagaceae	hrast, žir	14	N / C	AF, TL, CON, OT	ap, bch, fr, lf, sd, tr	building wood; furniture material; forest pasture for pigs (”žirovanje”); oak leaf as a motif for folk costume; acorns for reforestation; acorn cap as a whistle (“fučaljka”)	[22]
*Rhus typhina* L., Anacardiaceae, ZAGR79082	ruj	2	C	FO, OT	fr, ap	raw; decoration; juice	
*Ribes nigrum* L., Grossulariaceae, ZAGR79154	crni ribiz, crni ribiz divlji, crni ribizl, divlji ribizln	5	C	FO, DR	fr, sd	raw; syrup	[25]
*Ribes rubrum* L., Grossulariaceae, ZAGR79083	crveni ribizl, ribizl, ribiznl	5	C	FO, DR, AL	fr, sd	raw; liqueurs; jam	[25,26]
*Ribes uva-crispa* L., Grossulariaceae, ZAGR79130	ogrozd	7	C	FO	fr	raw	[24]
*Ribes* x *nidigrolaria* Rud.Bauer and A.Bauer, Grossulariaceae, ZAGR79135	josta	2	C	FO, DR	fr	raw; juice; jam	
*Ricinus communis* L., Euphorbiaceae	ricinus	4	C	COS, MD, OT	ap, fr, in, sd	hair growth oil (often used by men); anti-constipation oil used internally in very small amounts; decoration	[35]
*Robinia pseudoacacia* L., Fabaceae, ZAGR79084	agacija, bagrem, šemšir	26	N / C	FO, DR, AF, TL, MD, OT	ap, bch, fl, in, lf, tk, wh	raw; flower fried in batter or pancake mix and eaten as a wrap; nectar sucking; syrup; tea; good honey tree; hard garden stalks; firewood; leaf whistle; children’s trumpet “trubica”	[22,24,25]
*Rosa canina* L., Rosaceae, ZAGR79085	ruža, šipak	26	N	FO, DR, COS, MD	ap, bch, fl, fr, in, lf, st	raw; tea against diarrhea; jam; fragrance; decoration; for sale	[22,24,25,26,28,30]
*Rosmarinus officinalis* L., Lamiaceae, ZAGR79086	ružmarin	12	C	FO, DR, COS, TL, CE, MD, OT	ap, bch, fl, lf	condiment; tea for circulation; massage oil with olive oil against rheumatism; abortifacient (in large quantities); traditional wedding decoration	[22,25,26,30]
*Rubus caesius* L., Rosaceae, ZAGR79087	divlja kupina, šumska kupina	19	N	FO, DR, AL, MD, AF, OT	fr, fl, lf	leaf anti-diarrhea tea; jam; juice; syrup; wine; for distillation (“rakija”); remedy for weak calves	[22,24]
*Rubus idaeus* L., Rosaceae, ZAGR79131	divlja malina, šumska malina	5	N	FO, DR, AL	fr, lf	raw; jam; “himpersaft” juice (“malinovac”); tea; for distillation (“rakija”)	[22,24,25,26]
*Rubus vulgaris* Weihe et Nees, Rosaceae, ZAGR79088	kupina domaća	6	C	FO	fr	raw	[26]
*Rumex acetosa* L., Polygonaceae, ZAGR79089	kiselica, kiseljak, ščavljika,	25	N	FO, AF, MD	fl, fr, in, lf, st, ap	food snacking; salad; feed; tea against diarrhea for people and animals	[22,24,25,28]
*Rumex crispus* L., Polygonaceae, ZAGR79090	Štavljika, šćav, štavalj	2	N	FO, MD	lf, fl, ap	salad; tea against diarrhea for people and animals	[2]
*Rumex obtusifolius* L., Polygonaceae, ZAGR79091	čavlika, čavljika, poljska štavljika, ščavelj, ščavlika	5	N	AF, MD	bl, fl, fr, in	digestion agent for cattle; against diarrhea for people and animals	[2]
*Ruscus aculeatus* L., Asparagaceae, ZAGR79092	bodljikava veprina	3	N	CE, OT	ap	on cemeteries for All Saints Day; branches as decoration	[2,22,30]
*Ruscus hypoglossum* L., Asparagaceae, ZAGR79093	mekolisna veprina	2	N	OT	ap	decoration	
*Salix caprea* L., Salicaceae	cica maca, vrba	9	N	TL, CE, OT	bch, lf, ap	palms for Palm Sunday ceremony; weaving baskets after boiling in hot water; whistles; decoration	[25,43]
*Salix purpurea* L., Salicaceae	rakita	3	N	TL, OT	bch	weaving baskets; in the past: twigs for punishing children	[24,25]
*Salix* sp., *Salicaceae*	vrba	10	N / C	FO, TL, CON	ap, bch, lf	weaving baskets; kitchen utensils; water troughs; whistles	[25]
*Salvia officinalis* L., Lamiaceae	kadulja, ljekovita kadulja	2	N / C	AL, MD	fl, lf, in	tea against throat pain; liqueurs; massage oil; against moths and mosquitoes	[22,24,25,28,30]
*Salvia pratensis* L., Lamiaceae	divlja kadulja	2	N	COS, MD	lf, fl	tea against stomach pain	[33]
*Sambucus ebulus* L., Adoxaceae, ZAGR79094	abdovina, aftika, aptika	4	N	AL, MD	fl, fr, lf	in brandy; fruit to strengthen blood in horses; poultice from berries used for swelling and against rheumatism; antifungal treatment of nails	[24]
*Sambucus nigra* L., Viburnaceae, ZAGR79095	bazga	28	N	FO, DR, AL, TL, MD, OT	ap, bch, fl, fr, in	blossom fried in batter or pancake mix; flower or berry syrup; tea against cold and flu; syrup against cough: flower; milk and roasted sugar; in “rakija” brandy; tincture; wood for handles; shoe polish made of berries; toys from wood; for sale	[22,24,25,26,28]
*Secale cereale* L., Poaceae	raž	3	N	FO, AF, CE	fr, sd, ap	bread; animal feed; roof thatching	[24,25,26]
*Sempervivum tectorum* L., Crassulaceae, ZAGR79096	čuvarkuća, krovnjak	12	C	FO, DR, MD, OT	lf, pj	fresh salad; eaten mixed with honey for better immunity; ear infection or pain treatment (1—2 drops); ointment against swelling and warts; raw leaf treatments of wounds; poultice against infections; planted on the roof or in front of the house as decoration	[22,24,25,26,28]
*Silene vulgaris* (Moench) Garcke, Caryophyllaceae	škripavac	1	N	FO	lf	salad	[2,24,30]
*Silybum marianum* (L.) Gaertn., Asteraceae	sikavica	1	N	OT	in, sd	socialist cooperatives used to buy it out for the pharmaceutical industry	[24]
*Solanum lycopersicum* L., Solanaceae, ZAGR79097	paradajz	2	C	FO	fr	raw; cooked; pasteurized pureed juice	[25,26]
*Solanum lycopersicum* ‘Krumpiraš’, Solanaceae	paradajz ‘Krumpiraš’	1	C	FO	fr	cooking and preserves / pasteurized	
*Solanum lycopersicum* ‘Mađarac’, Solanaceae	paradajz ‘Mađarac’	1	C	FO	fr	raw; cooking	
*Solanum lycopersicum* ‘Volovsko srce’, Solanaceae, ZAGR79213	paradajz ‘Volovsko srce’	3	C	FO	fr	raw; cooking and preserves / pasteurized	
*Solanum tuberosum* L., Solanaceae, ZAGR79132	krumpir	6	C	FO, AF, MD	tb	cooking; for bread; small tubers with peel boiled for pigs; raw slices placed on face or soles against fever	[25,26]
*Sorbus domestica* L., Rosaceae	jabuka oskoruša, oskoruša	3	N / C	FO	fr	raw	[24,25]
*Sorbus torminalis* (L.) Crantz, Rosaceae	brekinja	2	N / C	FO, DR	fr	raw; for distillation (“rakija”)	[24]
*Sorghum bicolor* (L.) Moench, Poaceae	sirak	10	C	AF, TL	ap, bch, fr	animal feed; for making brooms	[26]
*Stellaria media* (L.) Vill., Caryophyllaceae, ZAGR79098	mišakinja, mišjakinja	4	N	FO, AF	ap, lf	fresh salad; pig and poultry feed	[33]
*Symphytum officinale* L., Boraginaceae, ZAGR79099	gavez	8	N	FOCOS, AL, AF, MD, OT	rt, fr, in, lf, sd	external use: poultice; massage tincture against rheumatism; root in “rakija” against arthritis; ointment with pig fat against skin diseases; arthritis; swelling; bumps; hemorrhoids; sciatica and sprains; for faster healing of wounds; fertilizer and pesticide	[22,24,25,26,28]
*Syringa vulgaris* L., Oleaceae, ZAGR79100	jorgovan	2	C	OT	bch, in, wh	gifted as a sign of sympathy or love; decoration	
*Tagetes patula* L., Asteraceae, ZAGR79101	kadifa, kadifica	2	C	MD, OT	fl, wh	against pests; decoration	[26]
*Tamus communis* L., Dioscoreaceae, ZAGR78924	bljušt, vilin korijen	2	N		sh, rt	prepared like asparagus; root in alcohol for poultice	[2,22]
*Tanacetum cinerariifolium* (Trevis.) Sch. Bip., Asteraceae	buhač	1	C	OT	wp	macerate sprayed against plant pests	[44]
*Taraxacum* spp., ZAGR79102	maslačak, divlji radič	25	N	FO, DR, TL, MD, OT, CE	ap, fl, in, rt, st	salad; dandelion “honey” (syrup extracted from flowers with sugar) against cough and to support immune system; root tea for blood pressure; as diuretic; whistle; ceremonial face wash on Palm Sunday	[22,24,25,26,28]
*Thymus serpyllum* L., Lamiaceae, ZAGR79103	majčina dušica, timijan	5	N / C	FO, DR, AL, MD	fl, lf	condiment; tea; liqueur	[22,24,26,28]
*Tilia cordata* Mill., Malvaceae, ZAGR79104	bijela lipa, lipa sitnolisna	25	N	FO, DRTL, CON, MD, OT	bch, fl, fr, in, lf, sd, st	tea against cold and flu; good for heart (but careful with heart patients!); tying material (“liko”) made from bark; kitchen utensils, building wood; furniture material; good honeybee pasture; toys: spinning tops; soltwood; easy to process	[26,28]
*Tilia platyphyllos* Scop., Malvaceae, ZAGR79105	crna lipa, rana lipa	3	N	DR, MD	fl, fr, lf	calming tea against asthma; cough	[22,24,25,28]
*Trapa natans* L., Lythraceae, ZAGR79106	bodljikavi orah, orašak, vodeni orah	4	N	FO	sd	boiled fruit for food; seeds for flour; collected for sale	[36]
*Trifolium incarnatum* L., Fabaceae	inkarnatka	2	C	FO, AF	lf, wh	young leaves used for salad; animal feed	
*Trifolium pratense* L., Fabaceae, ZAGR79107	divlja djetelina, djetelina	6	C	FO, AF, OT	ap, lf, in, st	animal feed; good honey plant; Easter eggs decorations used as a motif, game: looking for 4-leafed clover	[24,33]
*Triticum aestivum* L., Poaceae, ZAGR79133	pšenica	12	C	FO, AF, TL, CON, CE, MD, OT	ap, fl, fr, in, lf, sd, st	flour (bread); immature grain chewed as a snack; straw for animal feed; building material (chaff) for masonry ovens; ropemaking; starch glue (flour and water); straw traditionally placed under the Christmas tree	[24,25,26]
*Typha angustifolia* L., Typhaceae, ZAGR79108	rogoz	4	N	TL, MD, OT	fr, in, tr, wh	sealant between the boards; mosquito repellent; material for weaving baskets; decoration	[42]
*Urtica dioica* L., Urticaceae, ZAGR79109	kopriva	34	N	FO, DR, COS, AF, MD, OT	ap, lf, sd, st	bread and stew; green boiled salad; fried with eggs; detox tea; against rheumatism; boiled root for leg problems; boiled for poultry and pigs; tea or tincture to strengthen hair; for washing bottles; Easter eggs decoration; insecticide against aphids and Colorado beetles; organic fertilizer; dried for sale	[22,24,25,26,28]
*Valerianella locusta* (L.) Laterr., Valerianeaceae, ZAGR61064	divlji matovilac, matovilac, poljski matovilac	11	N / C	FO	ap, lf	salad; stew	[24,25,26]
*Vicia faba* L., Fabaceae, ZAGR79110	bob	5	C	FO	sd	cooked side dish; stew	[25,26]
*Viola odorata* L., Violaceae, ZAGR79111	ljubičica	2	N	MD, CE	fl	tea against throat problems; ceremonial face wash on Palm Sunday; for sale to pharmaceutical industry (against rheumatism)	[2,22,24,28,30]
*Viola tricolor*, Violaceae, ZAGR79431	mačuhica, ljubičica	6	N / C	FO, TL, OT	fl, lf, ap	edible flowers on salad; dried for tea; decoration	[33]
*Viscum album* L., Santalaceae, ZAGR79113	imela	4	N	FO, CE, MD, OT	lf, fr	tea for oncology patients; glue made from berries; Christmas tree decoration; a kiss under the mistletoe to bring eternal love	[22,24]
*Vitis labrusca* L., Vitaceae, ZAGR79114	loza ‘Otela’, loza ‘Bijela noja’, loza ‘Crna noja’, loza ‘Tudum’, loza ‘Nojem’	16	C	FO, DR, AL, OT	fr, lf	raw; wine; jam; for stuffed leaves with minced meat (“sarma”); leaves as toilet paper	
*Vitis vinifera* L., Vitaceae, ZAGR 79430	vinova loza, grožđe, vinova loza ‘Belovarca’	8	C	FO, AL, MD, OT	fr, in, lf, sd	raw; jam; vinegar; anti-hypertensive wine; for distillation (“rakija komovača”); best for massage; as insecticide; for stuffed leaves with minced meat (“sarma”)	[24,25,26]
*Vitis vinifera* ‘Bijeli delevar’, Vitaceae	vinova loza ‘Bijeli delevar’, vinova loza ‘Bijeli delevar’	2	C	AL, OT	fr, lf	wine; leaves as toilet paper; for stuffed leaves with minced meat (“sarma”)	
*Vitis vinifera* ‘Bijeli francuz’, Vitaceae	vinova loza ‘Bijeli francuz’	1	C	AL	fr	wine	
*Vitis vinifera* ‘Cardinal’, Vitaceae	vinova loza ‘Kardinal’	2	C	FO, DR, AL	fr	raw; wine	
*Vitis vinifera* ‘Chardonnay’, Vitaceae	vinova loza ‘Chardonay’	2	C	FO, AL	fr	raw; wine	[26]
*Vitis vinifera* ‘Crni francuz’, Vitaceae	vinova loza ‘Crni francuz’	1	C	AL	fr	wine	
*Vitis vinifera* ‘Francuz’, Vitaceae	vinova loza ‘Francuz’	2	C	FO, DR, AL	fr	raw; wine	
*Vitis vinifera* ‘Frankovka’, Vitaceae	vinova loza ‘Frankovka’	2	C	AL	fr	wine	
*Vitis vinifera* ‘Graševina’, Vitaceae	vinova loza ‘Graševina’	1	C	AL, OT	fr, lp, in	wine; leaves as a baking mat; for stuffed leaves with minced meat (“sarma”); “vinkot” cake: reduced must with condiments	[26]
*Vitis vinifera* ‘Hamburg’, Vitaceae	vinova loza ‘Hamburg’	1	C	FO, AL	fr, in	raw; wine	
*Vitis vinifera* ‘Izabela’, Vitaceae	vinova loza ‘Izabela’	9	C	FO, DR, AL, OT	fl, fr, lf, sd, st	raw; dried; wine; wine coloring; juice as sugar free syrup; for stuffed leaves with minced meat (“sarma”); leaves as toilet paper	[26]
*Vitis vinifera* ‘Kadarka’, Vitaceae	vinova loza ‘Kadarka’	4	C	FO, AL	fr, in	raw; wine	
*Vitis vinifera* ‘Kraljevina’, Vitaceae	vinova loza ‘Kraljevina’	1	C	FO, DR	fr	raw; wine	[26]
*Vitis vinifera* ‘Kraljica vinograda’, Vitaceae	vinova loza ‘Kraljica vinograda’	2	C	AL	fr	wine	
*Vitis vinifera* ‘Mirišavka’, Vitaceae	vinova loza ‘Mirišavka’	1	C	DR, AL	fr	wine; juice as sugar free syrup	
*Vitis vinifera* ‘Muškat žuti’, Vitaceae, ZAGR79215	vinova loza ‘Muškat žuti’	2	C	FO, AL, OT	fr, lf	raw; wine; dried; leaves as baking mat; for stuffed leaves with minced meat (“sarma”)	
*Vitis vinifera* ‘Muškat hamburg’, Vitaceae	vinova loza ‘Muškat hamburg’	2	C	FO, DR, AL	fr	raw; wine	[26]
*Vitis vinifera* ‘Nevenka’, Vitaceae	vinova loza ‘Nevenka’	1	C	AL, OT	fr, lf	wine; leaves as baking mat; for stuffed leaves with minced meat (“sarma”)	
*Vitis vinifera* ‘Pamid’, Vitaceae	vinova loza ‘Slankamenka’	2	C	FO, DR, AL	fr	raw; wine	
*Vitis vinifera* ‘Pinot’, Vitaceae, ZAGR79157	vinova loza ‘Crni pinot’ i ‘Bijeli pinot’	1	C	FO, AL	fr	raw; wine	[26]
*Vitis vinifera* ‘Plamenka’, Vitaceae	vinova loza ‘Plamenka’	1	C	FO, OT	fl, fr, lf	a motif for folk costumes; for stuffed leaves with minced meat (“sarma”)	
*Vitis vinifera* ‘Plemenka’, Vitaceae	vinova loza ‘Plemenka’	1	C	FO, AL	fr, lf	raw; wine; for stuffed leaves with minced meat (“sarma”)	
*Vitis vinifera* ‘Rajnski rizling’, Vitaceae	loza ‘Rizling’	1	C	FO, AL	fr	raw; wine; dried	
*Vitis vinifera* ‘Ružica’, Vitaceae	vinova loza ‘Ružica’	2	C	FO, AL	fr, in, lf	raw; wine; “vinkot” cake: reduced must with condiments; for stuffed leaves with minced meat (“sarma”)	
*Vitis vinifera* ‘Welschriesling’, Vitaceae, ZAGR79214	vinova loza ‘Graševina’, vinova loza ‘Talijanska graševina’	3	C	FO, DR, AL	fr	raw; wine	
*Vitis vinifera* ‘Zinfandel’, Vitaceae	vinova loza ‘Dalmatinac’	4	C	FO, AL, OT	fr, in, lf	raw; wine; dried; for stuffed leaves with minced meat (“sarma”); leaves as toilet paper	
*Wisteria sinensis* (Sims) Sweet, Fabaceae, ZAGR79115	glicinija	2	C	OT	fl, in, lf	decoration; plant attracts butterflies	
*Zea mays* L., Poaceae, ZAGR79116	kukuruz, kukuruz simplica	17	C	FO, DR, AL, AF, TL, MD, OT	ap, bk, bl, fl, fr, lf, rt, sd, st, tk	food (“polenta”, silk for tea against urological problems; grain for sauerkraut); animal feed “šlempa”: pomace with corn stalks; for distillation (ethyl alcohol); as toilet paper; toys; wicker baskets; slippers made of woven leaves; corn cobs: for children’s game “bunar” (well); “mlin” (mill); “školice” (school); for kindling; formerly mattress (“stroža”) filling	[25,26]
*Zea mays* ‘Osmoredac’, Poaceae	kukuruz ‘Osmoredac’	4	C	FO, TL, CE, MD, OT	fl, fr, st	boiled; corn cob drink against cold; hair curling tube; as toilet paper; game: water well	[24]
*Zea mays* ‘Pucanac’, Poaceae	kukuruz kokičar, kukuruz ‘Pucanac’	4	C	FO, TL	fr, in, lf, sd	roasted ripe corn; hair curling tube	
*Zea mays* ‘Stodanac’, Poaceae	kukuruz ‘Stodanac’	2	C	FO	fr, in, sd	boiled	
*Fungi*							
*Agaricus campestris* L., Agaricaceae	pečurka, šampinjon	9	N	FO	fruiting body	thermally processed	[24,45]
*Armillaria* spp., Physalacriaceae	mrazovača	2	N	FO	fruiting body	thermally processed	[26,45]
*Auricularia auricula-judae* (Bull.) Quél., Auricularaceae	petrovo uho, judino uho	3	N	FO	fruiting body	raw	[45]
*Boletus aereus* Bull., Boletaceae	hajdinski varganj, vrganj	2	N	FO	fruiting body	thermally processed; dried	[24,45]
*Boletus regius* Krombh., Boletaceae	kraljevka	1	N	FO	fruiting body	thermally processed; dried	[24,26]
*Boletus* sp., Boletaceae	varganj, vrganj	19	N	FO	fruiting body	thermally processed; dried; pickled	[24,26]
*Calocybe gambosa* (Fr.) Singer, Lyophyllaceae	đurđevača	1	N	FO	fruiting body	thermally processed	[45]
*Cantharellus cibarius* Fr. Cantharellaceae	lisičarka, žutica, lisičica, lisička	16	N	FO	fruiting body	thermally processed	[24,26,45]
*Coprinus comatus* (O.F. Müll.) Pers., Agaricaceae	gnojištarka, velika gnojištarka	1	N	FO	fruiting body	thermally processed	[45]
*Craterellus cornucopioides* (L.) Pers., Cantharellaceae	crna truba, mrka trubača, truba, trubača	4	N	FO	fruiting body	thermally processed; dried	[45]
*Entoloma clypeatum* (L.) P. Kumm., Entolomataceae	šljivarka, sivkasta šljivovača	4	N	FO	fruiting body	thermally processed	[45]
*Fomes fomentarius* (L.) Fr., Polyporaceae	guba	5	N	OT, DR	fruiting body	for tea; in bee smoker; to keep embers burning; for kindling	[45]
*Kuehneromyces mutabilis* (Schaeff.) Singer and A.H. Sm., Strophariaceae	mrazovača, panjevčica	3	N	FO	fruiting body	sautéed with onion; thermally processed	
*Lactarius deliciosus* (L.) Gray, Russulaceae	rujnica	1	N	FO	fruiting body	thermally processed	[45]
*Lactarius piperatus* (L.) Pers., Russulaceae	mleč, mlična, mliječak, mliječna, mlječica, mlječika, mlječka, mlječnica, mlječva	12	N	FO	fruiting body	thermally processed	[45]
*Langermannia gigantea* (Batsch) Rostk., Agaricaceae	divovska puhara, puhara	2	N	FO	fruiting body	thermally processed	[45]
*Leccinum aurantiacum* (Bull.) Gray, Boletaceae	turčin	1	N	FO	fruiting body	thermally processed	[45]
*Leccinum griseum* (Quél.) Singer, Boletaceae	brezovi varganj, pasji varganj, dedek, varganj kozjak	3	N	FO	fruiting body	thermally processed	[45]
*Lepista saeva* (Fr.) P.D. Orton, Tricholomataceae	mrazovača	5	N	FO	fruiting body	thermally processed	
*Lycoperdon perlatum* Pers., Agaricaceae and possibly other related species	prstić, pezduš, puhara, babin pušak	4	N	FO	fruiting body	thermally processed; raw	[45]
*Macrolepiota* spp., Agaricaceae	sunčanica	5	N	FO	fruiting body	thermally processed	[45]
*Marasmius oreades* (Bolton) Fr., Marasmiaceae	vilin klinčac, piličarka	1	N	FO	fruiting body	thermally processed	[45]
*Morchella* spp., Morchellaceae	smrčak	1	N	FO	fruiting body	thermally processed	[24,45]
*Pleurotus ostreatus* (Jacq.) P. Kumm., Pleurotaceae	bukovača	3	N	FO	fruiting body	thermally processed	[45]
*Pleurotus sapidus* Quél., Pleurotaceae	potpanjuška	1	N	FO	fruiting body	thermally processed	[45]
*Russula virescens* (Schaeff.) Fr., Russulaceae	golubica, zelena golubica	4	N	FO	fruiting body	thermally processed	[45]
*Sarcoscypha coccinea* (Jacq.) Lambotte, Sarcoscyphaceae	babino uho	3		FO	fruiting body	thermally processed; raw	[45]
*Suillus bovinus (L.)* Roussel, Suillaceae	kozji varganj	1	N	FO	fruiting body	thermally processed	

Abbreviations: Use categories: FO—dood, DR—drink, non-alcoholic, COS—cosmetics, AL—alcoholic drinks, AF—animal feed, TL—tools and utensils, CON—building and construction, CE—ceremonial use, MD—medicine, OT—other unspecified ways of use. Status: N—native or long-established species (archaeophyte), C—only cultivated, N/C– native and cultivated. Part used: ap—aerial parts, bch—branch, bd—bud, bk—bark, bl—bulb, fl—flowers, fr—fruit, hu—husk, hy—hypocotyl, im—immature fruit, in—inflorescence, lf—leaf, lg—legumes, pcl—pedicel, pe—petal, pj—plant juice, rh—rhizome, rs—resin, rt—root, sd—seed, sh—shoot, st—stalk, sty—styles, tb—tuber, tk—trunk, tr—tree, u—underground parts, wh—whole plant. Without literature numbers, the taxa recorded in this study, which have not yet been published in written regional sources (mostly these are new documented varieties of cultivated plants or taxa used as ornamental plants)

**Table 2 plants-13-02153-t002:** Fidelity levels (*FL*) for the ten taxa with the highest number of reports of use.

	Food	Drink, Non-Alcoholic	Cosmetics	Alcoholic Drinks	Animal Feed	Tools and Utensils	Medicine	Other Uses
*Urtica dioica* L.	0.50	0.28	0.19	0.00	0.33	0.00	0.14	0.36
*Robinia pseudoacacia* L.	0.72	0.34	0.00	0.00	0.03	0.41	0.03	0.21
*Sambucus nigra* L.	0.45	0.84	0.00	0.03	0.00	0.06	0.19	0.03
*Morus alba* L.	0.72	0.04	0.00	0.56	0.16	0.16	0.04	0.12
*Juglans regia* L.	0.61	0.00	0.00	0.43	0.00	0.17	0.30	0.35
*Rosa canina* L.	0.79	0.41	0.03	0.00	0.00	0.00	0.07	0.00
*Taraxacum* spp.	0.96	0.04	0.00	0.00	0.00	0.08	0.19	0.04
*Zea mays* L.	0.69	0.06	0.00	0.06	0.44	0.44	0.06	0.38
*Tilia* spp.	0.09	0.86	0.00	0.00	0.00	0.27	0.05	0.05
Mean	0.62	0.32	0.03	0.12	0.11	0.18	0.12	0.17
SD	0.23	0.32	0.06	0.20	0.16	0.16	0.09	0.15

Use categories ceremonial use and building or construction were not recorded for the listed taxa.

## Data Availability

The data presented in this study are available on request from the corresponding author.

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
