# Peer review of "Ethnobotany around the Virovitica Area in NW Slavonia (Continental Croatia)—Record of Rare Edible Use of Fungus Sarccoscypha coccinea"

_plants, 2024, doi:10.3390/plants13152153_

Round 1

Reviewer 1 Report

Comments and Suggestions for Authors

Ethnobotany Around the Virovitica Area in NW Slavonia (Continental Croatia) – Record of Rare Edible Use of Fungus

I have carefully read this work, which represents an exhaustive contribution to the ethnobotanical knowledge of a region of Croatia. In my experience, these studies are difficult to carry out since they are based on personal interviews, and many people are reluctant to answer the questions and not everyone knows the plants or their daily use. The number of plant and fungal species is quite high and the correct procedure for collection and subsequent analysis of the results has been followed.

To publish this work, I suggest that the following considerations be taken into account:

To better understand the results and discussion, in the methodology section, it should be put before, just after the introduction. This section should be expanded with climatological data from the study area, type of climate, hardiness zone, average altitude, etc. On the map you must include the icon of the North cardinal point and contextualize the place of study with another map in which you visually understand which part of the European Mediterranean region it corresponds to.

In the results and discussion section I would add a summarized phylogenetic tree, in which the most frequent families will appear; from a botanical point of view it is very clarifying.

In Table 1, ignore the reason why some taxa do not have herbarium numbers (Voucher), except for cultivated plant varieties, for example Vitis vinifera.

In the principal component analysis (PCA) (Figure 3) the statistical weight of each component is not indicated. Also, the figure must be enhanced to increase its sharpness)

According to my experience Pteridium aquilinum is very toxic, however it appears as edible.

Also in table 1 appears Quercus sp. What species can it refer to?

Author Response

REVIEWER 1

Ethnobotany Around the Virovitica Area in NW Slavonia (Continental Croatia) – Record of Rare Edible Use of Fungus

I have carefully read this work, which represents an exhaustive contribution to the ethnobotanical knowledge of a region of Croatia. In my experience, these studies are difficult to carry out since they are based on personal interviews, and many people are reluctant to answer the questions and not everyone knows the plants or their daily use. The number of plant and fungal species is quite high and the correct procedure for collection and subsequent analysis of the results has been followed.

  • Thank you for the comment, we are glad you like it.

To publish this work, I suggest that the following considerations be taken into account:

To better understand the results and discussion, in the methodology section, it should be put before, just after the introduction.

  • Thank you for the comment, it's not possible due to the magazine's policy, read the author's instructions, it has to stay that way. This strange order – methods at the end is the requirement of the journal.

This section should be expanded with climatological data from the study area, type of climate, hardiness zone, average altitude, etc.

  • Thank you for the comment, we added suggested text.

On the map you must include the icon of the North cardinal point and contextualize the place of study with another map in which you visually understand which part of the European Mediterranean region it corresponds to.

  • Thank you for your suggestion. A small map of the greater region showing the research area has been added as well as the North point. It is more informative now.

In the results and discussion section I would add a summarized phylogenetic tree, in which the most frequent families will appear; from a botanical point of view it is very clarifying.

  • Thank you for your suggestion, but this is not practiced in ethnobotanical publications.

In Table 1, ignore the reason why some taxa do not have herbarium numbers (Voucher), except for cultivated plant varieties, for example Vitis vinifera.

  • Thank you for your valuable comment, we have added a voucher to the herbarium of Vitis vinifera, ZAGR 79430.

In the principal component analysis (PCA) (Figure 3) the statistical weight of each component is not indicated. Also, the figure must be enhanced to increase its sharpness

  • Thank you for your valuable remark. The statistical weight of the two components is now indicated on the biplot. We also performed a new PCA in which we used the frequency of occurrence of a single plant in a particular use category as input data. We believe that we have obtained results that are more in line with real-life conditions, as the result is not only influenced by the fact that the plant is used for a specific purpose, but also by the frequency of use for a specific purpose.

According to my experience Pteridium aquilinum is very toxic, however it appears as edible.

  • Thank you for your valuable comment, Pteridium is eaten in many parts of the world after processing (e.g. China, Japan, Korea). Potato is also toxic without processing. We refer to a book on edible ferns which explains it in detail (Luczaj “Edible plants of the world”)

Also in table 1 appears Quercus sp. What species can it refer to?

  • Thank you for your valuable comment, we have added in Table 1. It can refer to several species primarily Quercus petrea and robur and to a lesser extent Quercus cerris.

Reviewer 2 Report

Comments and Suggestions for Authors

Article is curious, but it requires some attention before publication. My issues are listed belowe:

- Please carefully check all references, there are some minor mistakes, ex. line 391.

Author Response

Article is curious, but it requires some attention before publication. My issues are listed belowe:

- Please carefully check all references, there are some minor mistakes, ex. line 391.

  • Thank you for your valuable comment and opinion.
  • Thank you for noticing this. The problem seems to occur only in the pdf version of the document. In the MS Word version, everything is fine.

 In word version line 391 is completely ok, no mistakes; „no coprine was found in this species [51]. In Serbia C. comatus is used cooked or fried…. „ BUT in pdf version is [Error! Reference source not found.], on the place of [51].

Reviewer 3 Report

Comments and Suggestions for Authors

The manuscript is interesting but ill-conceived. Table 1 is very long, lacks order, lacks uniformity and NO reference to literature, especially in the case of medicinal use. Some plants are described to the species level, some to the variety level. Application, functional parts are difficult to read from this table.

Figure 1, 3, 4 species should be written in italics

PCA, what percentage of adjustment was component 1 and component 2?

Figure 1, 3 illegible

Figure 3, please mark the groups that are described, the graph looks slightly different than those described

There are a lot of self-citations, especially, among others, in the introduction. Łuczaj 18; Vitasović-Kosić 16 out of 62 references throughout the manuscript.

The manuscript is very long and if you add important information, it may be worth making it a book.

Author Response

The manuscript is interesting but ill-conceived. Table 1 is very long, lacks order, lacks uniformity and NO reference to literature, especially in the case of medicinal use. Some plants are described to the species level, some to the variety level. Application, functional parts are difficult to read from this table.

  • Thank you for your valuable comment. We preferred to keep table within the text and not put it in appendix to allow easier access for data. As this table only documents actual uses we did not see a need to add literature to it, especially that it is very long. We discussed some important species in the text.

Figure 1, 3, 4 species should be written in italics

  • Thank you for the remark. Despite the limitations of the software, we still found a way and printed the names in italics.

PCA, what percentage of adjustment was component 1 and component 2?

  • Thanks for the remark. The percentage has been added to the biplot.

Figure 1, 3 illegible

  • Thank you for your valuable suggestion. Your remark is well-founded, but due to the large amount of information and relatively small dimensions, the diagrams have a problem of illegibility. However, we have changed the graphics and we hope they are more readable now.

Figure 3, please mark the groups that are described, the graph looks slightly different than those described

  • Than you for the suggestion. While we were working on the response to your proposal, we decided to rerun the PCA so that the results would be closer to the actual plant utilisation distribution. The biplot and the PCA text were changed. Consequently, the text on PCA in the Methodology section has also been changed.

There are a lot of self-citations, especially, among others, in the introduction. Łuczaj 18; Vitasović-Kosić 16 out of 62 references throughout the manuscript.

  • Thank you for your valuable suggestion, unfortunately Luczaj and Vitasovic-Kosić were leaders of teams covering ethnobotanical studies in Croatia. We can only solve it by adding more references of other authors, but they do not exist for the area of ​​Croatia.

The manuscript is very long and if you add important information, it may be worth making it a book.

  • Thank you for your valuable suggestion, we are glad you like it, hopefull one day we can extend it, thank you for the remark.

Round 2

Reviewer 3 Report

Comments and Suggestions for Authors

Minor comments in the text have been taken into account, but the very extensive table has not been modified/ordered in any way to make it more transparent and legible. In addition, no sources of literature were added (Thirty-five people is small considering such extensive data obtained on many plants), especially when it concerns the use of individual plants in medicine, pharmacy, dietary supplements and the like.

The authors claim that there are many sources, and in terms of self-citations, that they are leaders in Croatia and there are few sources apart from self-citations...

Author Response

Dear Editors and Reviewer 3,

Thank you for further comments. Below is the answer to them.

Comment 1: Minor comments in the text have been taken into account, but the very extensive table has not been modified/ordered in any way to make it more transparent and legible

Response 1: 

The extensive table follows the format used in other ethnobotanical publications. One of the authors (Luczaj) published such long tables with results in many of his papers published in Plants, Journal of Ethnobiology and Ethnomedicine and Frontiers of Pharmacy. The reason to do so is to make sure that as much data as possible is contained in the main body of the text and not in attachments which may be overlooked or lost with time. Moreover the species mentioned work as keywords for search engines increasing the citations of the paper and the PLANTS journal – i.e. I strongly encourage the editors and reviewers to accept our concept of a long table. For comparison see similar table in other journals:

Nanagulyan, S., Zakaryan, N., Kartashyan, N., Piwowarczyk, R. and Łuczaj, Ł., 2020. Wild plants and fungi sold in the markets of Yerevan (Armenia). Journal of ethnobiology and ethnomedicine, 16, pp.1-27.

Łuczaj, Ł., Lamxay, V., Tongchan, K., Xayphakatsa, K., Phimmakong, K., Radavanh, S., Kanyasone, V., Pietras, M. and Karbarz, M., 2021. Wild food plants and fungi sold in the markets of Luang Prabang, Lao PDR. Journal of Ethnobiology and Ethnomedicine, 17, pp.1-27.

Ninčević Runjić, T., Jug-Dujaković, M., Runjić, M. and Łuczaj, Ł., 2024. Wild Edible Plants Used in Dalmatian Zagora (Croatia). Plants, 13(8), p.1079.

Łuczaj, Ł., Jug-Dujaković, M., Dolina, K., Jeričević, M., & Vitasović-Kosić, I. (2021). Insular pharmacopoeias: Ethnobotanical characteristics of medicinal plants used on the Adriatic islands. Frontiers in Pharmacology, 12, 623070.

Comment 2: In addition, no sources of literature were added (Thirty-five people is small considering such extensive data obtained on many plants), especially when it concerns the use of individual plants in medicine, pharmacy, dietary supplements and the like.

The authors claim that there are many sources, and in terms of self-citations, that they are leaders in Croatia and there are few sources apart from self-citations...

Response 2: 

We followed your comment and added more references of other authors to decrease the ratio of self-citations. We added 21 references. If you think we should add more, let us know.

In results/discussion we pointed out the most interesting findings and discussed them with literature WHEN we thought it was interesting new, but we avoided further prolonging a very long paper anyway but discussing more trivial findings.

We do not understand the comment “Thirty-five people is small considering such extensive data obtained on many plants”.

The number of interviews was not very high as the knowledge is undergoing strong devolution but we selected the most knowledgeable informants who could list and talk about dozens of species and varieties – i.e. they gave a lot of data in an area which was overlooked by ethnobotanical research before.

We enclose the 21 new references: 

  1.  Šugar, I., 2008. Hrvatski biljni imenoslov. Matica Hrvatska.

  1. Bakić, J. (1971). Neki pogledi preživljavanja na obalnom rubu Jadrana. Spasavanje ljudskih života na moru (naučne rasprave). In: Pomorska biblioteka, 23. Beograd: Izd. Mornaričkog glasnika, pp. 421–434

  1. Bakić, J. (1976). Upitnica – ‘Uporaba divljeg bilja i životinja u narodnoj prehrani’. Institut za pomorsku medicinu, Split.

  1. Pieroni, A. and Giusti, M.E., 2008. The remedies of the folk medicine of the Croatians living in Ćićarija, northern Istria. Collegium Antropologicum, 32(2), pp.623-627.

  1. Pieroni, A., Giusti, M.E., Münz, H., Lenzarini, C., Turković, G. and Turković, A., 2003. Ethnobotanical knowledge of the Istro-Romanians of Žejane in Croatia. Fitoterapia, 74(7-8), pp.710-719.

  1. Hu, S.Y., 2005. Food plants of China. Chinese University Press, Hongkong.

  1. Rasmussen, L.H., 2021. Presence of the carcinogen ptaquiloside in fern-based food products and traditional medicine: Four cases of human exposure. Current Research in Food Science, 4, pp.557-564.

  1. Boa ER. Wild edible fungi: a global overview of their use and importance to people. Food and Agriculture Organization of the United Nations

    1. Rexhepi B, Reka A. Ethno-mycological knowledge of some wild medicinal and food mushrooms from Osogovo Mountains (North Macedonia). Journal of Natural Sciences and Mathematics of UT. 2020 Nov 13;5(9-10):10-9.
